# DNA segment capture by Smc5/6 holocomplexes

Michael Taschner ● & Stephan Gruber ● ✉

Three distinct structural maintenance of chromosomes (SMC) complexes facilitate chromosome folding and segregation in eukaryotes, presumably by DNA loop extrusion. How SMCs interact with DNA to extrude loops is not well understood. Among the SMC complexes, Smc5/6 has dedicated roles in DNA repair and preventing a buildup of aberrant DNA junctions. In the present study, we describe the reconstitution of ATP-dependent DNA loading by yeast Smc5/6 rings. Loading strictly requires the Nse5/6 subcomplex which opens the kleisin neck gate. We show that plasmid molecules are topologically entrapped in the kleisin and two SMC subcompartments, but not in the full SMC compartment. This is explained by the SMC compartment holding a looped DNA segment and by kleisin locking it in place when passing between the two flanks of the loop for neck-gate closure. Related segment capture events may provide the power stroke in subsequent DNA extrusion steps, possibly also in other SMC complexes, thus providing a unifying principle for DNA loading and extrusion.

In eukaryotes, three distinct SMC complexes (cohesin, condensin and Smc5/6) share essential tasks in the maintenance of chromosome structure and the faithful transmission of genetic information during nuclear division (reviewed in ref. 1). Some of these ATP-powered, DNA-folding machines are known to form large DNA loops by loop extrusion[2–4]. This is thought to allow condensin to compact chromatids in mitosis and cohesin to participate in gene regulation, DNA repair and recombination in interphase. However, loop extrusion or merely DNA translocation may well be a conserved feature of all pro- and eukaryotic relatives. Smc5/6 has indeed recently been shown to extrude DNA loops or translocate along DNA in vitro[5], but the cellular functions and relevance of these activities remain unclear. DNA entrapment has long been considered a basic and essential feature of SMC function[1]. It provides a simple explanation for how SMC complexes stay in *cis* (on a given DNA molecule) over time, maintain directionality of translocation and readily bypass obstacles up to a few tens of nanometers in size that they encounter frequently on the chromosomal translocation track. However, the requirement of DNA entrapment and the exact nature of the DNA interaction remain contested, with all possibilities being considered in the recent literature (Fig. 1a): topological DNA entrapment,

DNA loop entrapment (also denoted as pseudo-topological entrapment) and exclusively external DNA association. The last scenario was bolstered by the perceived bypass of very large obstacles by purified cohesin and condensin in single-molecule imaging experiments[6]. Cryo-electron microscopy (cryo-EM) studies have not yet been able to adequately address this question and in particular little is known about how Smc5/6 associates with DNA. Resolving DNA topology will be vital to refute and refine models for how the SMC ATPase powers DNA translocation and loop extrusion[7,8].

Complete loss of Smc5/6 function leads to severe chromosome segregation defects in mitosis and meiosis[9–13]. Toxic DNA structures such as unresolved recombination intermediates, DNA intertwinings or incompletely replicated chromosomal regions prevent proper chromosome segregation in the absence of Smc5/6 (refs. 14–16). DNA translocation by Smc5/6 may be required to detect and eliminate such DNA junctions. The Smc5/6 holocomplex comprises eight subunits: Smc5/6 and Nse1-6 in *Saccharyomyces cerevisiae* (Extended Data Fig. 1a). All of them cause lethality when removed by gene disruption[10]. Smc5/6 and Nse4 together form the conserved ATPase core with an elongated shape that is characteristic for all SMC complexes[17–22]. Smc5 and Smc6

Department of Fundamental Microbiology, Faculty of Biology and Medicine, University of Lausanne, Lausanne, Switzerland.
✉e-mail: stephan.gruber@unil.ch

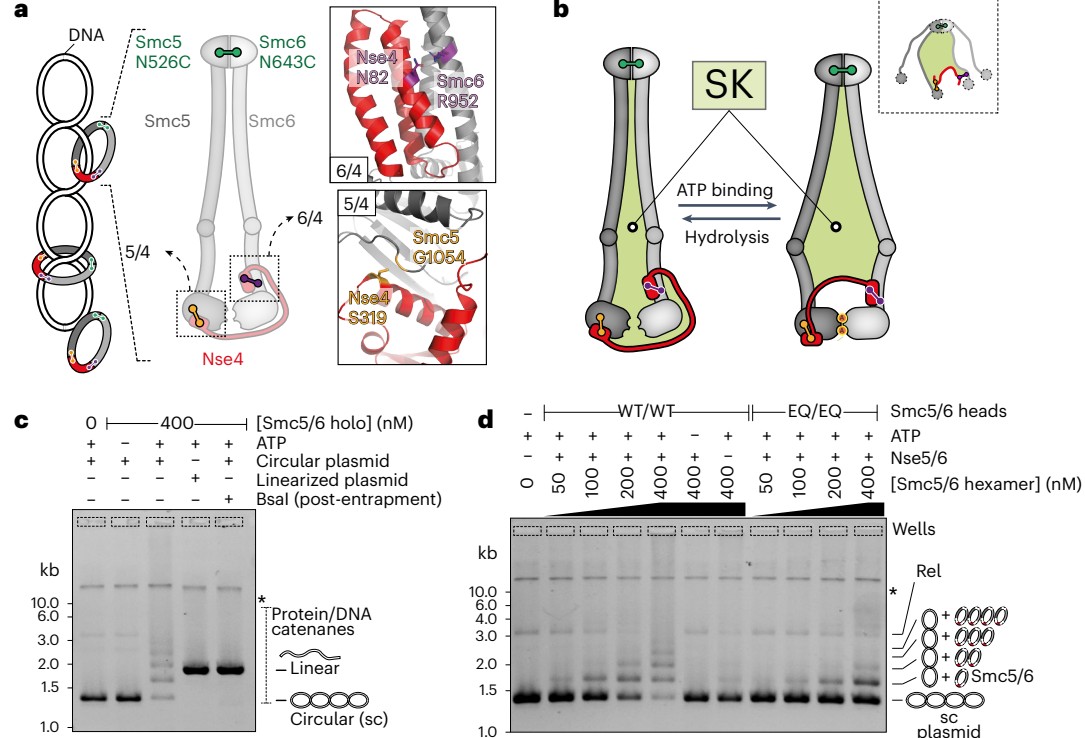

**Fig. 1 | DNA entrapment in Smc5/6. a,** Schematic view of a supercoiled circular DNA substrate (left) bound to SK rings (with topological DNA entrapment, top; nontopological DNA entrapment, middle; no DNA entrapment, bottom) and of the Smc5/6 ring subunits (middle). The positions of the cysteines that were engineered for crosslinking of the hinge, Smc6–Nse4 and Smc5–Nse4 interfaces are indicated in green, purple and orange, respectively. Close-up views of the positions of cysteines are shown at the Smc6–Nse4 (top right) and Smc5–Nse4 (bottom right) interfaces. **b,** Two schematic representations of the SK compartment depending on the positioning of the Nse4 kleisin subunit. Three cysteine pairs—necessary to covalently close it—are indicated with colored handlebars. A cartoon showing the crosslinked complex with the colored compartment after denaturation is shown in the dashed box. Positions of hinge as well as N- and C-terminal ATPase domains are still shown as half-ovals and circles, respectively. **c,** Coisolation of crosslinked Smc5/6 proteins with plasmid DNA by agarose gel electrophoresis. The results were obtained with an octameric Smc5/6 holocomplex harboring cysteine pairs for crosslinking of the SK ring, comparing linear and circular DNA substrates. Traces of contaminating linear DNA from *E. coli* are marked with an asterisk. BsaI, BsaI restriction endonuclease. **d,** As in **c** but with titration of the protein complex and comparison of entrapment efficiencies between the wild-type (WT) and ATP-hydrolysis-deficient (EQ/EQ) complexes. Controls without either ATP or the Nse5/6 loader are also included. The plasmid substrate is largely supercoiled (sc), but a relaxed (Rel) form is also visible.

proteins harbor the canonical 'hinge' domains for heterotypical dimerization connected via ~35-nm-long, antiparallel, coiled-coil 'arms' to globular ABC-type 'head' domains with highly conserved motifs for ATP binding and hydrolysis[23–25]. The kleisin subunit Nse4 bridges the head of the κ-SMC protein Smc5 to the head-proximal coiled coil ('neck') of the ν-SMC protein Smc6. This creates the SK ring structure shown in other SMC complexes to be, in principle, capable of entrapping DNA[26–30]. Nse4 serves as an attachment point for two kite proteins (the Nse1 and Nse3 subunits), also found in the prokaryotic SMC complexes[31], whereas the Smc5/6-specific subunit Nse2 attaches to the coiled-coil arm of Smc5 (ref. 32). Finally, a stable subcomplex is formed by the Nse5 and Nse6 subunits[19,21]. It contacts the Smc5/6 hexamer via multiple interfaces, including one on the arms and one on the heads. Single-molecule tracking and biochemical studies suggest a function of Nse5/6 in chromosome loading[21,33]. In *S. pombe* (but not in *S. cerevisiae*), disruption of *nse5* and *nse6* genes cause less severe phenotypes when compared with deletion of other Smc5/6 subunits, indicating that the hexamer can partially function without Nse5/6 in fission yeast[34]. Curiously, loop extrusion by *S. cerevisiae* Smc5/6 does not depend on the loader Nse5/6 and is actually inhibited by it, possibly implying that Smc5/6 holocomplexes have functions beyond loop extrusion[5]. A dedicated loader appears to be involved in viral restriction by Smc5/6 in human cells[35].

The ATP hydrolysis cycle involves major structural rearrangements in the SMC complex[36–41]. Three main conformations have been delineated for Smc5/6 (Extended Data Fig. 1b)[17–22]: (1) in the 'ATP-engaged

state', two molecules of ATP are sandwiched by residues of the Walker A and B motifs of the Smc5 head and the signature motif of the Smc6 head and vice versa; the engaged heads keep the two head-proximal arms at a distance, thus opening an SMC compartment[36,42,43]; (2) in the 'juxtaposed state', the heads are ATP disengaged and the Smc5 and Smc6 arms coalign, yielding a closed SMC compartment[22]; and (3) the 'inhibited state' relies on the intercalation of the loader Nse5/6 between the arms and heads of Smc5/6 (refs. 20,21), thus preventing head engagement and ATP hydrolysis. The inhibition is, in turn, overcome by the binding of a suitable DNA substrate[21]. A head–DNA interface on top of the engaged heads was described for several SMC complexes[38,44–50]. Such a DNA-clamping state has recently also been described for Smc5/6 by cryo-EM[43].

In the present study, we demonstrate that Smc5/6 holocomplexes efficiently entrap DNA molecules. Using the reconstituted DNA-loading reaction, we elucidate the position and topology of DNA in the Smc5/6 complex and unambiguously identify the neck gate as the entry gate for plasmid DNA.

## Results

### Topological DNA entrapment by the Smc5/6 ring

Salt-stable DNA binding by the octameric yeast Smc5/6 holocomplex depends on a circular DNA substrate, being indicative of a topological component of the Smc5/6 DNA association[21]. In the present study, we delineate the position of DNA in the Smc5/6 complex by creating covalently closed DNA compartments followed by protein–DNA coisolation

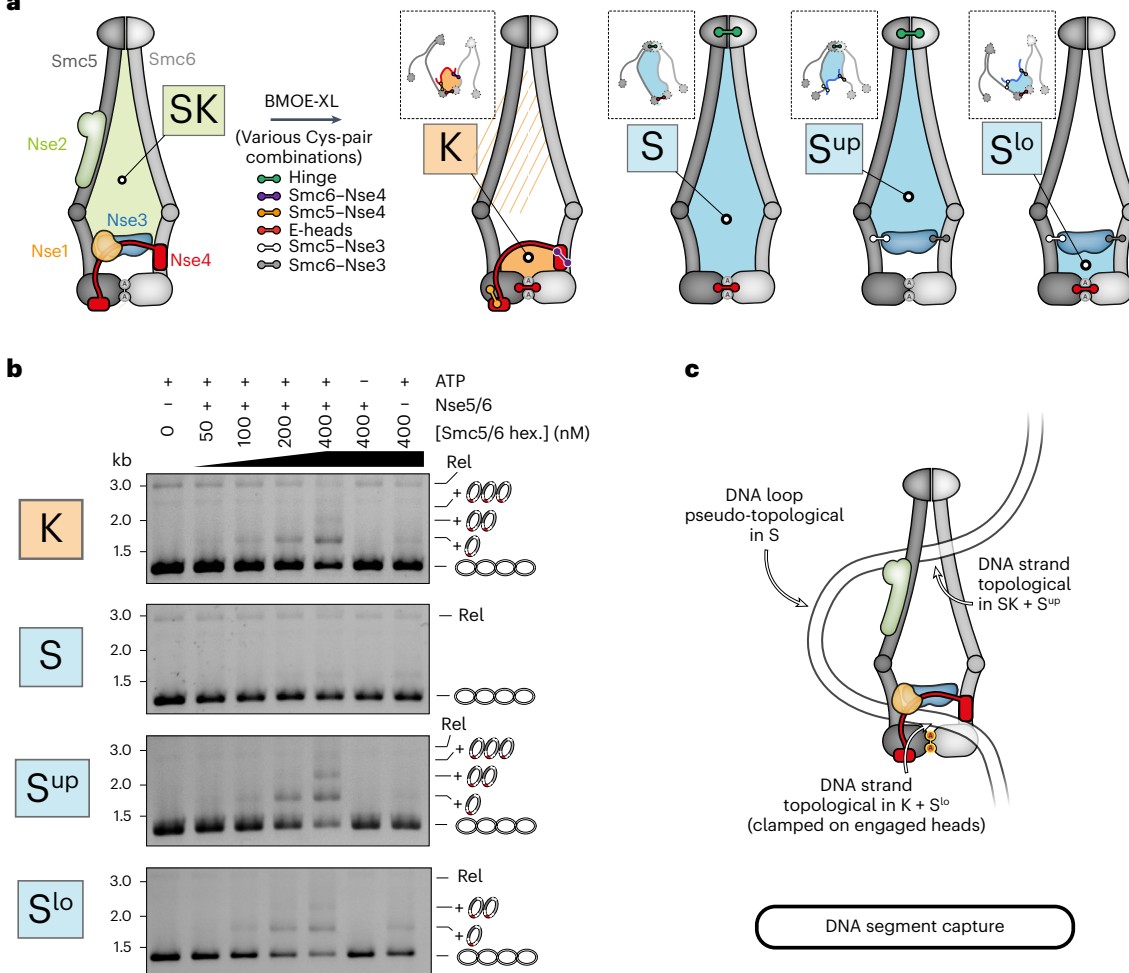

**Fig. 2 | DNA entrapment in Smc5/6 subcompartments. a**, Schematic representation of subcompartments formed during ATP-dependent head engagement. The scheme on the left shows the complete Smc5/6 hexamer with the SK compartment highlighted as in Fig. 1. Combinations of cysteine pairs (colored handlebars as indicated) lead to covalent closure of K and S compartments, with the latter being split into upper (S^up) and lower (S^lo) compartments by Nse3 crosslinking as indicated. Note that the schemes on the right show only the crosslinked subunits that remain attached after denaturation. Schemes in dashed boxes indicate compartments after denaturation as in Fig. 1b. XL, Crosslink; E, Engaged. **b**, Coisolation of crosslinked Smc5/6 proteins with plasmid DNA by agarose gel electrophoresis. Results obtained with protein preparations harboring cysteine pairs for crosslinking of the K (top), S (second top), S^up (third top) or S^lo (bottom) compartments, as in Fig. 1d. Hex, Hexamer. **c**, Schematic representation of a DNA segment-capture state explaining the findings in **b**.

under protein-denaturing conditions (Fig. 1a). First, we tested whether the perimeter of the Smc5/6 ring entraps plasmid DNA by crosslinking the three ring interfaces to generate a covalently closed SK ring. We used structure prediction (by AlphaFold-Multimer) to position cysteine pairs at two ring interfaces (Fig. 1a)[51,52]. We observed robust chemical crosslinking of the Smc5–Nse4 interface (~80%) and the Smc6–Nse4 interface (~90%) on addition of bis-maleimidoethane (BMOE; Extended Data Fig. 2a, lanes 3 and 4), as shown for the Smc5/6 hinge[21]. When cysteine pairs at all three ring interfaces were combined, crosslinking gave rise to a substantial fraction of covalently closed Smc5/6 SK ring species (Extended Data Fig. 2a, lane 8 and Fig. 1b). The cysteine mutations did not have strong effects on the Smc5/6 ATPase activity (Extended Data Fig. 2b).

We then separated protein–DNA catenanes from free DNA species by agarose gel electrophoresis (schematics in Extended Data Fig. 3a)[53]. Crosslinking at the three SK ring interfaces resulted in a characteristic laddering of a small (1.8-kb) plasmid at an elevated Smc5/6 concentration (400 nM) (Fig. 1c), presumably due to the reduced mobility of DNA molecules associated with increasing numbers of Smc5/6 complexes. The laddering was fully dependent on ATP and disappeared when plasmid DNA was linearized by a restriction enzyme before gel

electrophoresis, as expected for a topological interaction (Fig. 1c). This loading reaction reached a steady state relatively quickly (within minutes; see below). Moreover, three protein preparations lacking one of the six engineered cysteines showed little or no DNA laddering, with the residual gel shift probably explained by low levels of off-target crosslinking (Extended Data Fig. 2c).

The laddering was virtually abolished in the absence of the loader Nse5/6 (Fig. 1d) and also clearly reduced by ATP-hydrolysis-deficient mutations in both Smc5 and Smc6 ('EQ/EQ'), indicating that ATP hydrolysis promotes more robust DNA entrapment. Based on similar experiments using protein gel electrophoresis, we estimate that a large fraction of Smc5/6 entraps plasmid DNA under our experimental conditions (Extended Data Fig. 2d). We conclude that topological DNA loading by the Smc5/6 holocomplex is a robust and efficient reaction.

## DNA entrapment in the kleisin compartment

We next wondered where DNA might be located in the Smc5/6 complex. We previously showed that, under the conditions promoting salt-stable DNA binding and DNA entrapment (Fig. 1c–d), the complex exhibits head engagement as measured by cysteine crosslinking[21]. Head engagement produces an SMC ('S') and a kleisin ('K') compartment

(see schemes in Fig. 2a). DNA must be in at least one of the two compartments. We first combined the cysteine pair at ATP-engaged heads with cysteines at both SMC–Nse4 interfaces to generate a covalently closed K compartment, which yielded robust DNA entrapment again in an ATP- and loader-dependent manner (Fig. 2b, top). The efficiency of entrapment in the K compartment appeared somewhat reduced when compared with the SK ring, but this is probably explained by the reduced efficiency of engaged-head crosslinking (~25%)[21]. A series of samples crosslinked at different time points after mixing showed that entrapment was detectable after 30 s and was saturated within a few minutes in both the SK ring and the K compartment (Extended Data Fig. 2e), supporting the notion that the two types of entrapment emerge from the same biochemical reaction and possibly correspond to the same state. We conclude that the K compartment of the ATP-engaged Smc5/6 holocomplex is occupied by DNA.

## DNA segment capture in the SMC compartment

Next, we combined head and hinge cysteines to create a covalently closed S compartment (Fig. 2a). Contrary to the K compartment, no loader-dependent DNA entrapment was observed for the S compartment (Fig. 2b, second panel). This may suggest that the S compartment is devoid of DNA when heads are ATP engaged. Alternatively, a DNA loop (rather than a single DNA double helix) may thread into the S compartment with the pseudo-topologically held DNA being lost on protein denaturation even after crosslinking. To avoid confusion with the to-be-extruded DNA loop, we designate this configuration as DNA segment capture (Fig. 2c)[54]. To test for DNA segment capture, we sought to split the S compartment into two subcompartments that each entrap a single DNA double helix rather than a DNA loop. A cryo-EM structure showed that the Nse3 subunit may split the S compartment into halves by contacting the Smc5 as well as the Smc6 coiled coil in a DNA-clamping state (Protein Data Bank (PDB), accession no. 7TVE)[43]. We generated cysteine pairs to crosslink Nse3 to both Smc5 and Smc6 (Extended Data Fig. 4) and combined them with hinge cysteines to create an upper SMC compartment (S$^{up}$) and with the head cysteines to generate a lower SMC compartment (S$^{lo}$) (Fig. 2a). Intriguingly, the cysteine combinations for the S$^{up}$ and the S$^{lo}$ compartments both resulted in robust DNA laddering (Fig. 2b, lower panels), demonstrating that a DNA segment is efficiently captured in the SMC compartment and held as a loop with (at least) one DNA passage in the S$^{lo}$ compartment and another one (or more) in the S$^{up}$ compartment (Fig. 2c).

## The loader Nse5/6 opens the neck gate in Smc5/6

In the course of the above experiments, we noticed that the crosslinking of the cysteine pair at the Smc6–Nse4 interface, designated the neck gate, is strongly sensitive to ligands. Although crosslinking was robust (~90%) with or without ATP and DNA in the absence of the loader (Fig. 3, lanes 2–5), it was strongly hampered when the loader Nse5/6 was added, especially when plasmid DNA was present (Fig. 3, lanes 7 and 8). Such a behavior was not observed for cysteine pairs at the hinge[21], along the coiled-coil arms (Extended Data Fig. 5) or at the Smc5–Nse4 interface (Extended Data Fig. 6a). Of note, addition of the loader led to some off-target crosslinking of a native cysteine in Nse5 with Smc5 (G1054C) (Extended Data Fig. 6b). More importantly, however, neck-gate crosslinking in the octamer was partially restored by addition of ATP alone (Fig. 3, lane 9) and virtually fully restored by addition of both ATP and DNA (lane 10). Similar crosslinking efficiencies were observed with complexes harboring all cysteines for SK ring crosslinking (Extended Data Fig. 6c). Experiments performed with complexes carrying mutations of active site residues showed that this stimulatory effect of DNA on neck-gate closure requires ATP engagement of Smc5/6 heads (Extended Data Fig. 6d, lanes 9 and 10) but not ATP hydrolysis (Extended Data Fig. 6e, lanes 9 and 10). Altogether, the presented results thus suggest that binding of the Smc5/6 hexamer to the loader Nse5/6 specifically detaches one of the three ring interfaces.

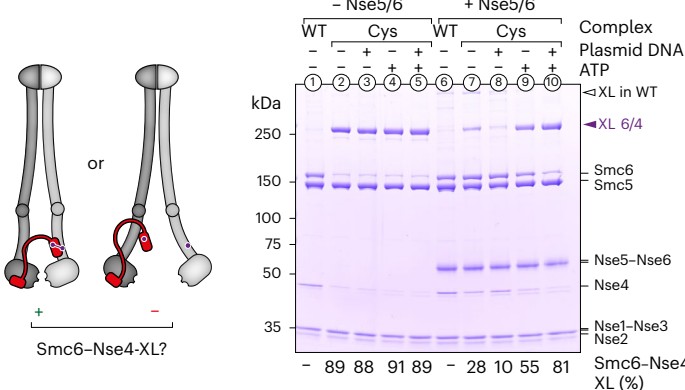

**Fig. 3 | Opening of the neck gate by Nse5/6.** Crosslinking of purified Smc5/6 hexamers with cysteines at the Smc6–Nse4 interface is shown in the presence and absence of ligands. Detection of crosslinked species was by SDS–PAGE and Coomassie staining. Loss of crosslinking suggests that the gate was open. XL, Crosslink. Equivalent experiments for this interface containing ATPase head mutations as well as for other cysteine pairs are shown in Extended Data Fig. 6.

DNA clamping, that is, ATP engagement of SMC heads, together with DNA binding at the head–DNA interface, appears to promote gate closure. It is interesting that fusion of Nse4 to Smc6, but not to Smc5, is lethal in *S. cerevisiae*, supporting the notion that opening of the neck gate is essential for Smc5/6 function (Extended Data Fig. 6f)[18]. The neck gate is largely closed in the presence of ATP and DNA, implying that, once loaded onto DNA, Smc5/6 holocomplexes form a ring that encompasses the DNA double helix.

## DNA clamping is required for efficient gate closure

The above observations suggested that DNA binding plays a role in neck-gate closure. We next determined whether the head–DNA interface is indeed crucial for topological DNA entrapment or important only subsequently, for example, for the conversion of a putative 'DNA-holding' intermediate into the DNA segment-capture state (see below). Based on sequence conservation and structure comparison (Extended Data Fig. 7a–c), we identified three positively charged residues on Smc5 (K89, R139, R143) and another three on Smc6 (R135, K200, K201) as putative DNA-binding residues (Fig. 4a). To test whether these residues are important for DNA entrapment, we mutated them by alanine and glutamate substitution (charge removal and reversal, respectively) in isolation or in combination. Mutant alleles were tested for functional complementation of a respective deletion mutant in yeast using plasmid shuffling[55]. *Smc5* alleles harboring double or triple glutamate substitutions resulted in a strong growth phenotype, whereas alanine mutations were apparently tolerated well (Fig. 4b). The *smc6* gene was somewhat more sensitive to the mutagenesis with the triple alanine mutant also exhibiting clear growth retardation (Fig. 4c). A selection of residues based on a recent cryo-EM map[43] resulted in similar outcomes (Extended Data Fig. 7d,e). The deficiencies of the single, double and triple alanine mutants in *smc6* were aggravated when combined with the *smc5(3A)* allele, suggesting that these residues have partially overlapping functions (Extended Data Fig. 7f) as previously observed for bacterial Smc[38]. The sextuple alanine mutant strain *(smc5(3A), smc6(3A)* or 3A3A) displayed a null-like phenotype, supporting the notion that the head–DNA interface is crucial for an essential function of Smc5/6.

Selecting one partially and one fully defective mutant (*smc6(3A)* and 3A3A, respectively) we tested for DNA loading of Smc5/6 when DNA binding at the heads is compromised. First, we measured ATP-dependent, salt-stable DNA binding by pull-down assays as described previously[21] (Extended Data Fig. 3b). The Smc6(3A) variant

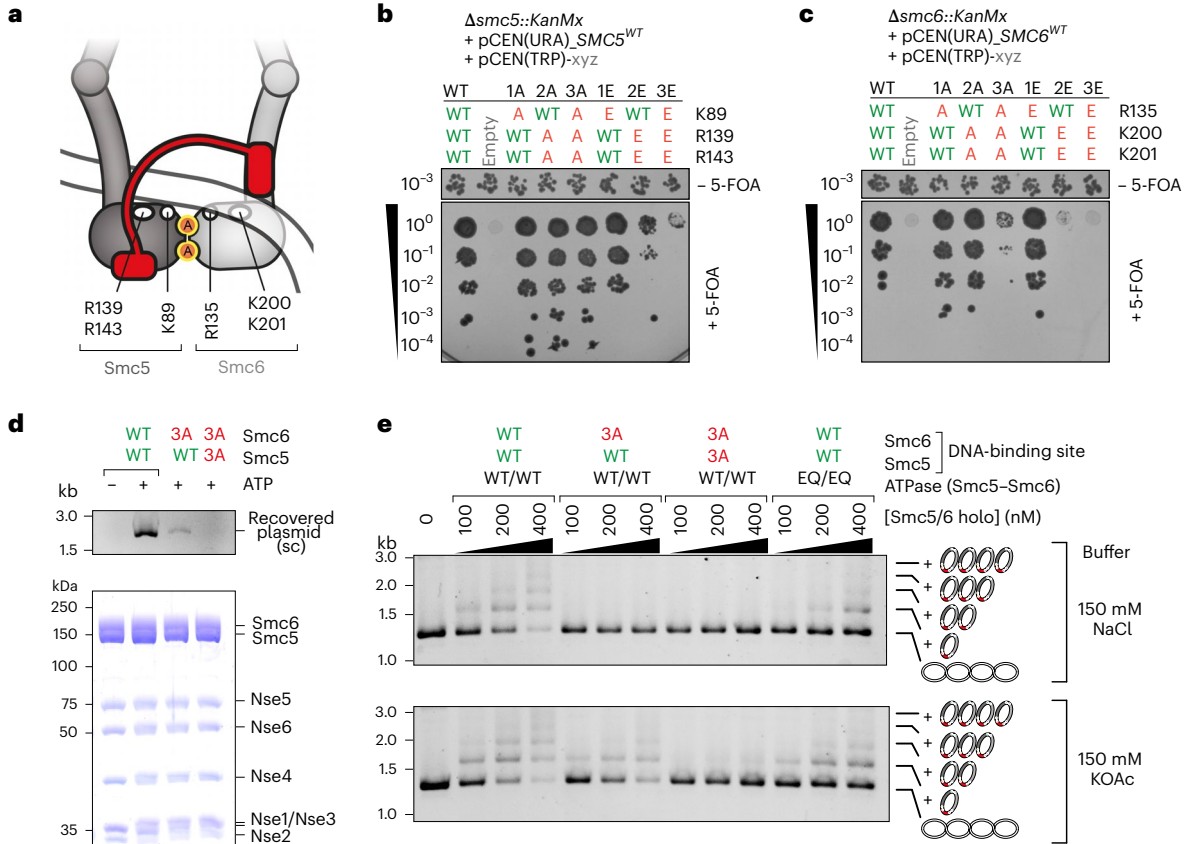

**Fig. 4 | DNA clamping is essential for DNA entrapment.** Identification of putative DNA-binding residues in Smc5 and Smc6. **a**, The positions of the selected residues schematically displayed for the DNA-clamping state (Extended Data Fig. 7a). **b**, Positively charged residues on the Smc5 head mutated to alanine (A) or glutamate (E) in isolation or in combination as indicated. The mutant alleles were tested for function by plasmid shuffling. Counterselection against a pCEN(URA3) plasmid carrying a wild-type (WT) *SMC5* allele by addition of 5-FOA revealed *smc5* mutants resulting in a growth defect. **c**, As in **b** for residues in Smc6.

**d**, Salt-stable DNA binding with wild-type and mutant Smc5/6 as determined by protein immobilization using a TwinStrep tag on Smc6 (bottom) and detection of coisolated plasmid DNA ('recovered plasmid') by agarose gel electrophoresis (top). **e**, DNA entrapment in the SK ring of wild-type, DNA-binding and ATP-hydrolysis-defective Smc5/6 mutants. Standard DNA entrapment salt conditions (150 mM NaCl) were used as in Fig. 1d (top) or buffer with reduced ionic strength (150 mM KOAc) as for the salt-stable DNA-binding assay in **b**.

only poorly recovered plasmid DNA, whereas the 3A3A variant was completely unable to do so, together strongly suggesting that DNA binding is crucial for Nse5/6-mediated DNA loading (Fig. 4d). Next, we combined the Smc6(3A) and the 3A3A mutations with the cysteine variants for SK crosslinking and DNA entrapment. Using the standard buffer conditions (150 mM NaCl), both mutants failed to support any DNA entrapment (Fig. 4e, top). When the reaction buffer was adjusted to mimic the conditions used for salt-stable DNA binding (150 mM KOAc; a 'milder' salt with larger ions), the Smc6(3A) variant supported residual DNA entrapment, whereas the 3A3A variant did not (Fig. 4e, bottom). These results demonstrate that the head–DNA binding interface is crucial for topological DNA entrapment by Smc5/6. Of note, the mutant complexes showed slightly reduced ATPase activity (Extended Data Fig. 2b). However, the defect in DNA entrapment is not explained by the lack of ATP hydrolysis, because the EQ/EQ complex quite efficiently entrapped DNA despite failing to support any noticeable ATPase activity (Fig. 4e and Extended Data Fig. 2b). Moreover, the mutant complexes produced ring crosslinking albeit at somewhat reduced efficiency, in particular the 3A3A mutant, an effect that was more pronounced in the stringent salt buffer (Extended Data Fig. 6g). The latter is probably explained by defects in neck-gate closure, because the complex lacking DNA-binding residues on Smc5 and Smc6 heads also failed to display DNA-stimulated neck-gate closure (Extended Data Fig. 6h). DNA binding at the heads might thus serve (at least) two

related purposes during DNA entrapment: recruiting DNA for entrapment and promoting gate closure.

## DNA passage through the neck gate

We wondered whether DNA might indeed enter the Smc5/6 ring by passing through the neck gate that opens on association with the loader Nse5/6 and closes on DNA encounter. To test this hypothesis directly, we created an Smc6–Nse4 fusion protein. The linker peptide covalently connects the carboxy terminus of the Smc6 polypeptide to the amino terminus of Nse4 and includes a recognition sequence for cleavage by the 3C protease (Extended Data Fig. 8a). This construct is nonfunctional in vivo (Extended Data Fig. 6f; as previously reported for a related construct[18]). The reconstituted Smc5/6 complex harboring this fusion protein hydrolyzed ATP normally (Extended Data Fig. 2b). We next combined the Smc6–Nse4 fusion protein with cysteines for DNA entrapment experiments (see schematics in Extended Data Fig. 8b). When combining cysteines at the hinge and the Smc5–Nse4 interface with the Smc6–Nse4 fusion protein, the ring species failed to produce any gel shift, demonstrating that DNA entrapment is not possible when Smc6 is linked to Nse4, as expected if the neck gate indeed serves as the DNA entry gate (Fig. 5a, lanes 5–7 and Extended Data Fig. 8b). Curiously, however, we recovered DNA laddering when combining the Smc6–Nse4 fusion with only the neck-gate crosslink albeit with an altered laddering pattern and distinctively smaller steps, probably

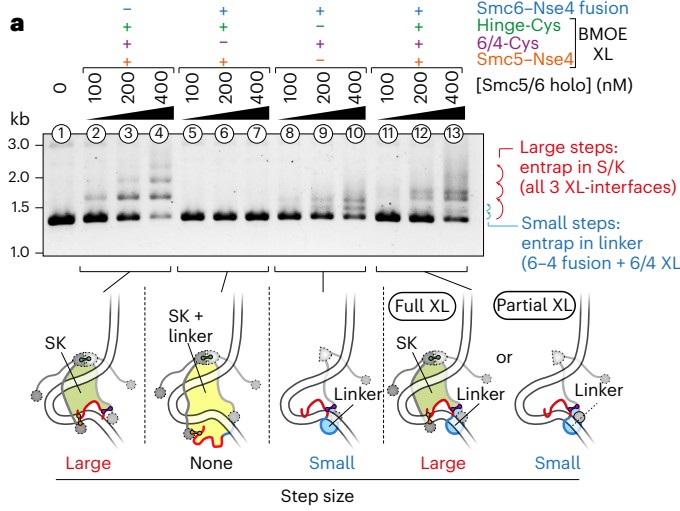

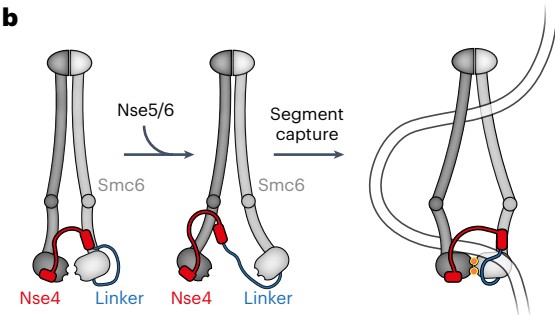

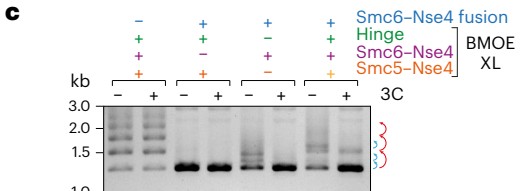

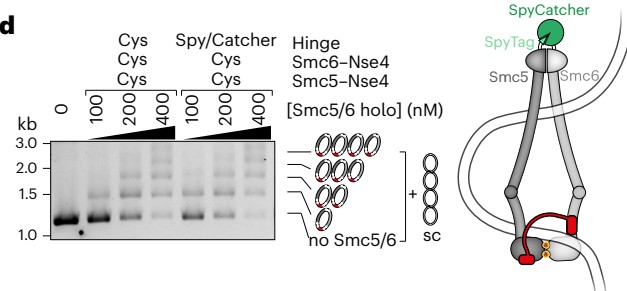

**Fig. 5 | DNA passes through the Smc6–Nse4 gate. a**, Coisolation of crosslinked Smc5/6 proteins with plasmid DNA by agarose gel electrophoresis. Results were obtained with protein preparations containing combinations of cysteine pairs and an Smc6–Nse4 fusion protein. Schematic drawings of crosslinked and denatured protein species are shown below the gel, with the closed compartments indicated. Different observed step sizes of DNA ladders are caused by differences in the size of the crosslinked protein species. XL, Crosslink. **b**, Schematic representation of DNA clamping in the presence of an Smc6–Nse4 fusion protein. The length of the linker allows it to wrap around the DNA strand clamped on ATP-engaged heads. **c**, As in **a** with Smc5/6 holocomplexes at 400 nM, and with or without post-entrapment opening of the linker using the human rhinovirus 3C protease. **d**, Coisolation of crosslinked Smc5/6 proteins with plasmid DNA by agarose gel electrophoresis, comparing a protein preparation harboring cysteine pairs for crosslinking of the SK ring with one in which the hinge cysteine pair is replaced by a SpyTag–SpyCatcher fusion. Topological DNA entrapment in Smc5/6 is not prevented by the hinge fusion, unlike what is shown for cohesin.

owing to the smaller size of the protein ring species formed with Smc6 and Nse4 protein only (Fig. 5a, lanes 8–10 and Extended Data Fig. 8b). Combining the Smc6–Nse4 fusion with all three cysteine pairs yielded DNA laddering with larger as well as smaller laddering steps, presumably due to partial crosslinking (Fig. 5a, lanes 11–13 and Extended Data Fig. 8b). To reconcile these observations, we suggest that the linker may be long enough to embrace the incoming DNA, thus not blocking the loading reaction itself (Fig. 5b). This would lead to pseudo-topological entrapment of a DNA loop in an expanded 'SK-plus-linker' compartment (see schematics of closed and denatured compartments in Fig. 5a), thus explaining the absence of laddering when the Smc6–Nse4 interface was closed only by the peptide linker but not by the crosslink. Cleavage of the linker peptide by 3C protease before (Extended Data Fig. 8c) or after (Fig. 5c) the loading reaction eliminated the entrapment by the linker compartment and also removed the smaller-sized steps observed with the fusion protein plus all three cysteine pairs. Also consistent with the linker wrapping around DNA, we find that the Smc6–Nse4 fusion complex supported normal or near-normal, salt-stable DNA binding regardless of whether the peptide linker was left intact or cleaved, whereas a control complex with an open kleisin did not (Extended Data Fig. 8d). Taken together, these results strongly suggest that the engineered peptide linker indeed embraces the incoming DNA and identifies the neck gate as the DNA entry gate.

Our experiments imply that all plasmid DNA loading happens via the neck gate. However, a recent study showed that, in addition to the neck gate, the hinge serves as DNA entry gate for topological entrapment by cohesin in vitro[53]. Related experiments performed with the covalently linked Smc5/6 hinge domains (using a SpyTag–SpyCatcher pair; Extended Data Fig. 8e) did not display any noticeable DNA entrapment defects (Fig. 5d), confirming that passage via the hinge does not substantially contribute to DNA entrapment in Smc5/6 and highlighting intriguing differences in DNA loading between these SMC complexes.

## The Smc5/6 ring entraps plasmid DNA in vivo

Finally, we wanted to investigate whether DNA entrapment by the Smc5/6 complex also occurs in vivo. We introduced the cysteine pairs for SK ring crosslinking into the *nse4*, *smc5* and *smc6* genes in budding yeast by allelic replacement. After BMOE crosslinking in cells, immunoprecipitation and southern blotting analysis, as developed to detect cohesin DNA coentrapment[27], we could clearly demonstrate the entrapment of a CEN plasmid by Smc5/6, similar to cohesin but without leading to cohesion of plasmid DNA (Fig. 6). A control strain lacking one of the six cysteines for crosslinking of Smc5/6 did not show DNA entrapment.

## Discussion
### DNA entrapment by SMC complexes
Many of the diverse SMC functions in genome maintenance and chromosome organization are thought to arise directly from a DNA motor activity[1]. Revealing the DNA topology underlying loop extrusion and translocation is a prerequisite for a basic understanding of SMC activity. We show in the present study that purified preparations of Smc5/6 holocomplexes quickly and effectively entrap circular DNA substrates in a topological manner. This reaction is strictly ATP and loader dependent, and DNA entrapment is also detectable in vivo. We reveal the DNA entry gate and characterize the trajectory of DNA into the Smc5/6 ring, showing that DNA loading involves a looped DNA substrate and eventually leads to the capture of a DNA segment by Smc5/6.

Although the notion of DNA entrapment was challenged again recently by single-molecule imaging, which apparently showed the bypass of very large obstacles on DNA by translocating cohesin and condensin[6], we argue based on our findings and previous reports that DNA entrapment must not be discounted. Earlier studies measuring salt-stable DNA binding are in support of DNA entrapment by cohesin, condensin and Smc5/6, but they were short of revealing the

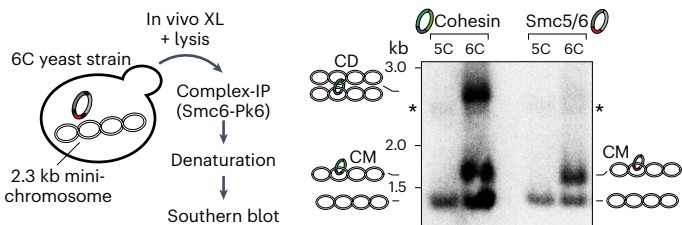

**Fig. 6 | Smc5/6 entraps DNA in vivo.** The scheme on the left shows an outline of the procedure. Results from southern blotting on the right show that crosslinkable versions (6C) of both cohesin and Smc5/6 entrap the mini-chromosome (catenated monomers (CM)), but only cohesin does so in a cohesive manner (catenated dimers (CD)). Control strains lacking one of the ring cysteines (5C) abolish entrapment in both cases.

mode of DNA entrapment or the DNA-containing compartment[17,21,56–58]. Experiments with entire bacterial chromosomes have provided strong evidence for DNA entrapment in the SK ring (and the K but not the S compartment) of Smc-ScpAB and MukBEF but did not reveal the mode of entrapment due to the complexity of the DNA substrate (including branches on the replicating chromosome)[28,41]. Cohesin SK rings are known to support DNA entrapment in vivo at least in the context of sister chromatid cohesion[27]. Single-molecule imaging experiments with cohesin, however, suggest that loop extrusion might not require cohesin ring opening, implying entrapment of a DNA loop or external DNA association rather than topological DNA entrapment[2]. Entrapment of a DNA loop has recently also been proposed for the SK ring of yeast condensin based on coisolation experiments after site-specific crosslinking[50].

Our data clearly demonstrate the topological DNA entrapment by Smc5/6 in vivo and elucidate its reliance on Nse5/6 in vitro. Nse5/6 has previously been implicated in chromosome loading in vivo by single-molecule tracking in fission yeast[33]. Curiously, Nse5/6 is dispensable for DNA loop extrusion by Smc5/6 in vitro and actually hinders it[5]. To reconcile these observations, one may invoke Nse5/6 converting dynamic loop-extruding Smc5/6 complexes into more stably bound DNA-entrapping complexes. The latter may (or may not) support DNA translocation[54]. The functions of all these states and activities remain to be discerned.

### The loader opens the neck gate for DNA entry

Strict topological entrapment (as previously documented for cohesin and in the present paper for Smc5/6) requires the passage of one annular particle through an opening in another. In the present study, we unequivocally demonstrate that the neck gate serves as the major and probably the only entry gate for plasmid DNA in Smc5/6. This conclusion is based on an Smc6–Nse4 fusion protein, in which the linker does not block DNA loading but entraps the incoming DNA in an artificial linker compartment. The identity of the entry gate is furthermore supported by its efficient opening on contact of the Smc5/6 hexamer with the loader Nse5/6 and by the lethality caused by the relevant fusion in vivo in budding yeast[18]. Other possible gates do not seem to contribute to DNA entrapment in the reconstituted reaction (as judged by the absence of DNA entrapment by complexes harboring the Smc6–Nse4 fusion protein (as well as cysteines for crosslinking the hinge and Smc5–Nse4)). The situation appears more complicated in other SMC complexes. The neck gate is known to support DNA exit in cohesin[59–63] and has also been implicated in DNA entry in cohesin[48,58] as well as condensin[64]. A recent paper demonstrated that cohesin DNA entry can be supported by the hinge and the neck gate in vitro, however, with entry only via the hinge being dependent on the loader protein Scc2 (ref. 53), possibly suggesting that the hinge is the main or physiological gate for DNA entry, in contrast to what we observed in the present study for Smc5/6. Moreover, sister chromatid cohesion can be built by cohesin

complexes harboring an Smc3–Scc1 'neck-gate' fusion protein. This fusion protein also supports DNA entrapment by cohesin in vivo (when combined with cysteines for crosslinking of the other ring interfaces, unlike observed with Smc5/6 in Fig. 5)[27,65,66], again implying fundamental differences with Smc5/6. A recent study using fusion proteins, in contrast to earlier reports, suggested that condensin loading does not require an entry gate, at least for DNA loop extrusion[50].

We find that the association with the loader destabilizes the contact between Smc6 and Nse4. Knowing that the loader intercalates between Smc5 and Smc6 heads[20,21] and that the Nse4 middle part is rather short, we propose that steric constraints prevent Nse4 from being attached to both SMC proteins simultaneously when the loader is also bound, resulting in the detachment of the least stable interface. DNA binding might reverse the effect of the loader, by evicting it from between the SMC heads and enabling productive head engagement (and ATP hydrolysis)[21] as well as ring re-closure by Smc6–Nse4 association. Assuming that the presence of DNA keeps the neck gate shut also during subsequent ATP hydrolysis cycles, this would result in a stable DNA association possibly facilitating DNA translocation over extended periods of time. The neck-gate interface also appears labile in purified preparations of cohesin and condensin (sub)complexes with the binding partners detaching from one another on ATP-head engagement[64,67] (or incubation with cohesin unloading factors[68]). A similar reaction has been proposed for fission yeast Smc5/6 based on yeast two-hybrid experiments[69]. Neck-gate opening on ATP binding and head engagement, however, diametrically contrasts with our finding on budding yeast Smc5/6, where ATP-dependent head engagement leads to closure rather than opening of the gate (Fig. 7). Whether these mechanistic differences have biological consequences will be important to work out.

### DNA segment capture for loading as well as translocation

We propose that DNA loading involves DNA segment capture as an essential intermediate. We find that head–DNA binding is required for gate closure as well as for DNA entrapment. The head–DNA contacts conceivably hold a bent DNA segment in place during loading and further DNA contacts guide the K subunit around DNA to ensure entrapment of a DNA double helix in the SK ring (Fig. 2c). Of note, a model proposed for cohesin DNA loading via the neck gate resembles our notion of DNA segment capture[48]. However, DNA loading via the cohesin neck gate has since been shown to work regardless of the absence or presence of the loading–clamping subunit Scc2, thus challenging this notion[53].

Why would DNA loading of SMC complexes involve such an intricate structure rather than rely on entrapping a single DNA double helix? DNA segment capture has previously been proposed to explain DNA translocation and DNA loop extrusion by bacterial Smc-ScpAB and other SMC complexes[36,54] (Fig. 7). We envisage that the segment-capture states during loading and translocation/loop extrusion are structurally related. The notion of DNA segment capture mediating DNA loading as well as translocation explains how the very same ATPase core supports two distinct biochemical reactions and thus provides a unified framework for SMC ATPase functions, possibly also applicable to other SMC-related processes. The segment-capture state is consistent with observations on the bacterial SMC complexes Smc-ScpAB and Muk-BEF made by in vivo crosslinking and chromosome isolation, as well as a cryo-EM structure of the MukBEF complex encompassing two separate DNA molecules[38,41]. If true, the similarities in DNA topology are remarkable, especially when considering the large evolutionary distances across the three SMC complexes. Notably, the entrapment results were obtained with cysteines reporting distinct ATPase states (that is, predominantly the juxtaposed and the ATP-engaged state for Smc-ScpAB and Smc5/6, respectively; in the case of MukB the cysteines did not strongly discriminate between the states), indicating that DNA topology does not change after initial loading and the entry gate may remain shut during subsequent ATP hydrolysis cycles (Fig. 7).

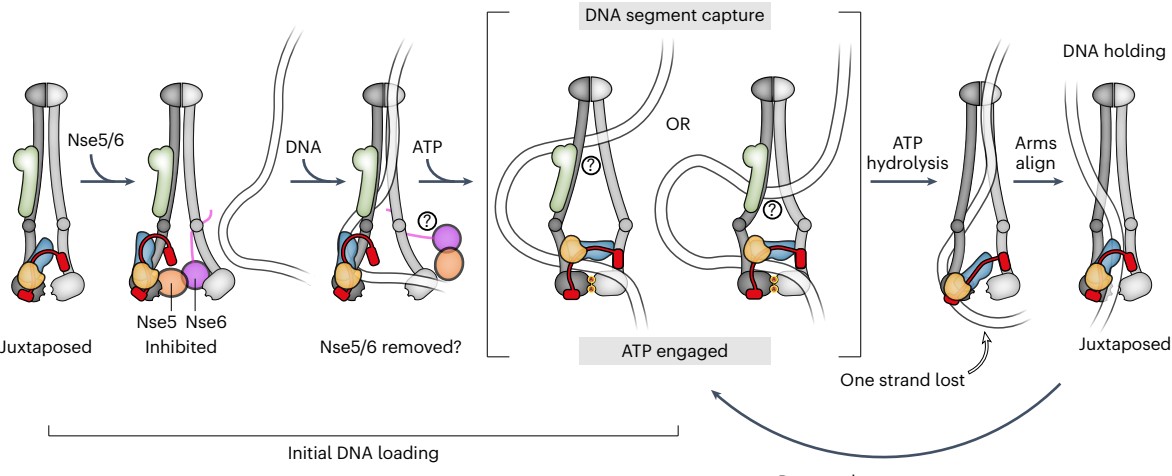

**Fig. 7 | Model for chromosome loading and DNA translocation by alternating between a holding and a segment-capture state.** Schematic representation of the main findings presented in the present study. The segment-capture state (middle) is an essential intermediate in both DNA loading (left) and DNA translocation (right). Note that two possibilities for this state are indicated, differing in their degree of arm opening toward the hinge.

Recent data suggest that loop extrusion by condensin does not require the opening of a dedicated DNA entry gate and may therefore not involve the same loading intermediate[50]. Subsequent steps of loop extrusion, however, could still proceed via DNA segment capture[50,54].

Future experiments will have to elucidate how exactly DNA entrapment by Smc5/6 is linked to its function and its activities as translocator and loop extruder. Although a segment capture-type mechanism explains our findings, other suitable scenarios may exist or emerge in the future.

## Online content

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

## Methods

### Cloning of expression plasmids

Plasmids containing multiple expression cassettes for subunits of the *S. cerevisiae* Smc5/6 complex were cloned as described in Taschner et al.[21]. Transcription of Smc5-containing cassettes was regulated with a tac-promoter and lambda-terminator, whereas the other subunits were transcribed by T7 promoters and terminators. A list of all expression plasmids used in the present study can be found in Supplementary Table 1.

### Protein expression and purification

All proteins and protein complexes described in the present study were expressed in *Escherichia coli* (DE3) Rosetta transformed with either a single plasmid or a combination of two plasmids. Supplementary Table 2 lists plasmid details about all described complexes. For all purifications, 1 liter of the strain carrying the desired plasmid(s) was grown in Terrific Broth medium at 37 °C to an optical density at 600 nm ($OD_{600}$) of 1.0 and the culture temperature was reduced to 22 °C. Expression was then induced with isopropyl β-ᴅ-1-thiogalactopyranoside at a final concentration of 0.4 mM and allowed to proceed overnight (typically for 16 h). All Smc5/6 complexes and the Nse5/6 dimer were purified following a published procedure described in detail in Taschner et al.[21].

### Cloning of yeast plasmids

Coding sequences for wild-type or mutant Smc6, Smc5 or Nse4 were cloned between their respective endogenous upstream and downstream regulatory sequences by Golden Gate cloning into centromeric acceptor plasmids pCEN(URA) or pCEN(TRP). N- or C-terminal tags were added as indicated. For plasmids containing two loci, we combined the individual cassettes by an additional Golden Gate Assembly step. A list of all yeast plasmids used in the present study can be found in Supplementary Table 3.

### Creation of yeast strains

All strains in the present study were created on the W303 background. Functional assays to investigate mutants of Smc6, Smc5 and Nse4 were carried out using plasmid shuffling[55]. To generate suitable strains, we first deleted the respective loci in a diploid strain then transformed the resulting strain with a pCEN(URA) plasmid containing the wild-type locus. Sporulation of this strain allowed us to isolate haploid offspring with the locus on the pCEN(URA) as the sole source for the respective protein. For double shuffling strains, both loci were deleted in the diploid and re-introduced on the pCEN(URA) plasmid. Haploid shuffling strains were transformed with pCEN(TRP) plasmids containing either a wild-type or a mutant version of the respective loci. A list of all yeast strains used in the present study can be found in Supplementary Table 4.

### Plasmid shuffling assays

Haploid shuffling strains containing the wild-type locus on pCEN(URA) and another wild-type or mutant locus on pCEN(TRP) were grown on plates lacking uracil and tryptophan. Single colonies were inoculated in minimal medium lacking only tryptophan and grown for 18–24 h at 30 °C. Four tenfold dilutions were then prepared in water and 2 µl of the five samples (undiluted culture and the four dilutions) were spotted on one plate lacking uracil and tryptophan, and on another containing all amino acids as well as 1 mg ml⁻¹ of 5-fluorouracil (5-FOA) (selecting for cells that had maintained or lost the pCEN(URA) plasmid, respectively). Plates were incubated at 30 °C and pictures were taken at suitable time points (between 36 h and 60 h) to score growth of strains containing mutant versions of the respective Smc5–6 components.

### Site-specific BMOE crosslinking in vitro

Smc5/6 hexamers with or without indicated cysteines were diluted to a final concentration of 0.5 µM in ATPase buffer (10 mM Hepes-KOH,

pH 7.5, 150 mM KOAc, 2 mM $MgCl_2$ and 20% (v:v) glycerol) in a total volume of 30 µl. In reactions containing the Nse5/6 dimer, this complex was added in a 1.25× molar excess (0.625 µM). For reactions containing ATP and/or plasmid DNA (25 kbp), these ligands were added at a final concentration of 2 mM and 5 nM, respectively. The circular or linear plasmid substrate was produced as described in Taschner et al.[21]. Protein and substrates were incubated for 5 min at room temperature (RT) after mixing and BMOE was then added at a final concentration of 1 mM. After 45 s of incubation, dithiothreitol (DTT) was added at a final concentration of 10 mM to stop the reaction. For post-crosslinking treatment with benzonase, 1 µl of the concentrated enzyme stock (750 U µl⁻¹) was added to the tube and incubated for 15 min at RT. Samples were mixed with sodium dodecylsulfate (SDS) gel-loading dye, heated to 80 °C for 15 min and then analyzed by polyacrylamide gel analysis on Novex WedgeWell 4–12% (w:v) Tris-glycine gels or 3–8% (w:v) Tris-acetate gels (Invitrogen). Tris-glycine gels were run at 180 V for 1 h at RT and Tris-acetate gels at 4 °C for 3 h at 30 mA. Gels were fixed for 1–2 h in gel-fixing solution (50% (v:v) ethanol and 10% (v:v) acetic acid) and stained overnight using Coomassie staining solution (50% (v:v) methanol, 10% (v:v) acetic acid and 1 mg ml⁻¹ of Coomassie Brilliant Blue R-250). Gels were destained in destaining solution (50% (v:v) methanol and 10% (v:v) acetic acid) and then rehydrated and stored in 5% (v:v) acetic acid. Quantification of bands in scanned gel images was done using Fiji[70].

### Analysis of salt-stable DNA association

These assays were carried out as described in Taschner et al.[21] with modifications; 100 µl reactions in ATPase buffer (10 mM Hepes-KOH, pH 7.5, 150 mM KOAc, 2 mM $MgCl_2$ and 20% (v:v) glycerol) were set up containing combinations of the following components at the indicated concentrations: Smc5/6 hexameric complex (wild-type or mutant) 600 nM, Nse5/6 complex 900 nM, nucleotide 2 mM and 3 µg of plasmid (pSG4418, 2.8 kbp). After incubation for 10 min at RT, 500 µl of ice-cold, high-salt buffer (20 mM Tris, pH 7.5, and 1,000 mM NaCl) was added and the mixture was incubated with 20 µl of StrepTactin Sepharose HP (GE Healthcare) for 45 min to pull out proteins via a C-terminal TwinStrep tag on Smc6. Beads were harvested by centrifugation (700g, 2 min) and washed twice with 1 ml of high-salt buffer. The bound material was then eluted with a buffer containing 20 mM Tris, pH 7.5, 250 mM NaCl and 5 mM desthiobiotin. Aliquots of the eluate were supplemented with either 6× gel-loading dye containing SDS (Thermo Fisher Scientific), heated to 65° for 10 min and analyzed by agarose gel electrophoresis (1% (w:v) agarose in 0.5× TBE (Tris/borate/EDTA)) to check the DNA content, or 2× SDS gel-loading dye, heated to 95 °C for 10 min and analyzed by SDS–polyacrylamide gel electrophoresis (PAGE) to visualize eluted proteins.

### Topological DNA-binding assays

Analysis of topological DNA binding was performed following a protocol described in Collier et al.[49] with the following modifications. For clear detection of 'DNA laddering', a small plasmid (1,800 bp, pSG6085) was used at a final concentration of 15 nM and incubated with indicated concentrations of proteins in the presence or absence of 1 mM ATP. After incubation at RT (for 2 min except in the case of the time course shown in Fig. 2c), BMOE was added to a final concentration of 1 mM. Crosslinking was allowed to proceed for 30 s at RT and then stopped by the addition of 10 mM DTT. For post-XL treatment with 3C protease, 1 µl of a stock of the home-made enzyme (concentration around 1 mg ml⁻¹) was added to the tube and incubated for 20 min at RT. The samples were supplemented with 6× loading dye with SDS and heated for 20 min at 70 °C. Then, 10 µl aliquots were separated on 1% (w:v) agarose gels containing 0.03% (w:v) SDS and 1 µg ml⁻¹ of ethidium bromide. Gels were run at RT for 3 h at 5 V cm⁻¹.

### Mini-chromosome entrapment assays

Entrapment of mini-chromosomes in vivo in budding yeast was performed following a procedure previously used for cohesin[66].

**Article**

Briefly, strains expressing a crosslinkable version ('6C') of cohesin or Smc5/6 and containing a TRP mini-chromosome were grown together with control strains lacking one of the ring cysteines ('5C') overnight in medium lacking tryptophan. When the cultures reached an $OD_{600}$ of around 0.5, 40 ml (20 ODs) of the cultures was harvested by centrifugation (3,000 r.p.m., 3 min), and the pellets were washed once with 25 ml ice-cold phosphate-buffered saline (PBS) and subsequently transferred to screw-cap tubes. Pellets were then resuspended in 500 ml of ice-cold PBS; 30 µl of a 150 µM BMOE solution in dimethyl sulfoxide was added and crosslinking was allowed to proceed for 6 min on ice. The cells were then pelleted, the supernatant was discarded and cell pellets were snap-frozen in liquid nitrogen for storage at −80 °C. For lysis, 700 µl of lysis buffer (50 mM Hepes-KOH, pH 7.5, 100 mM KCl, 0.05% (v:v) Triton X-100, 0.025% (v:v) NP-40, 10 mM Na citrate and 25 mM Na sulfite), freshly supplemented with a cOmplete protease inhibitor tablet (Roche) and 1 mM phenylmethylsulfonyl fluoride, was added to the pellet together with acid-washed glass beads (425–600 µm), and the cells were opened using bead beating with a FastPrep 24 system (MPBio; 3×1 min with 5-min breaks on ice). The lysate was clarified by centrifugation at 14,000$g$ for 10 min at 4 °C, incubated with mouse monoclonal anti-V5 antibody (BioRad) for 90 min at 4 °C to bind the C-terminal Pk6-epitope on Smc6 and complexes were then captured with Dyna-Beads Protein-G for another 90 min at 4 °C. Beads were washed twice with 1 ml of wash buffer (10 mM Tris-HCl, pH 8.0, 250 mM LiCl, 0.5% (v:v) NP-40, 0.5% (w:v) Na deoxycholate and 1 mM EDTA) and once with 1 ml of TE (Tris and EDTA), and bound material was then eluted with 1× DNA-loading dye in TE supplemented with 1% (w:v) SDS and preheated to 65 °C. The obtained material was separated on a 0.8% (w:v) agarose gel (in 1× TAE (TE + acetic acid)) overnight at 4 °C and 1.5 V cm$^{-1}$, and DNA was then transferred to Hybond-N$^+$ membranes by southern blotting using alkaline transfer. Mini-chromosomes were detected with a probe against the TRP marker, which was labeled with [α-$^{32}$P]ATP using the Prime-It II Random Primer Labeling Kit (Agilent) according to the manufacturer's instructions. Membranes were exposed to a Phosphor Storage Screen (Fujifilm) and signals were detected with a Typhoon Scanner (Cytiva).

### ATPase assays

Analysis of ATPase activity of selected Smc5/6 complexes was carried out exactly as described in Taschner et al.[21].

### Statistics and reproducibility

Determination of ATP hydrolysis rates was performed in three technical replicates. Mean and s.d. values were calculated. No data were excluded from the analyses. Qualitative assays of DNA entrapment were reproduced at least three times from independently grown cultures or purified material with comparable outcomes.

### Reporting summary

Further information on research design is available in the Nature Portfolio Reporting Summary linked to this article.

## Data availability

The following publicly available datasets were used as a guide for site-directed mutagenesis: PDB accession nos. 6ZZ6 and 7TVE. Source data are provided with this paper.

## Code availability

The present paper does not report any unique code.

## References

70. Schindelin, J. et al. Fiji: an open-source platform for biological-image analysis. *Nat. Methods* **9**, 676–682 (2012).

## Acknowledgements

We thank J. Collier, K. Nasmyth and M. Srinivasan for guidance and sharing reagents for the plasmid coentrapment experiments. We thank S. Pelet and the Pelet lab for advice and for materials and reagents for genetic engineering in yeast, and F. Bürmann, M. Räschle and members of the Gruber lab for comments on the manuscript and helpful discussions. This work was supported by the Swiss National Science Foundation (grant no. 310030L_170242) and the European Research Council (grant no. Horizon 2020 ERC CoG 724482) to S.G.

## Author contributions

S.G. conceived the project and acquired the funding. M.T. provided the methodology and investigations. S.G. and M.T. wrote the paper.

## Funding

## Competing interests

The authors declare no competing interests.

## Additional information

**Extended data** is available for this paper at https://doi.org/10.1038/s41594-023-00956-2.

**Correspondence and requests for materials** should be addressed to Stephan Gruber.

**Peer review information** *Nature Structural & Molecular Biology* thanks the anonymous reviewers for their contribution to the peer review of this work. Peer reviewer reports are available. Sara Osman and Dimitris Typas were the primary editors on this article and managed its editorial process and peer review in collaboration with the rest of the editorial team.

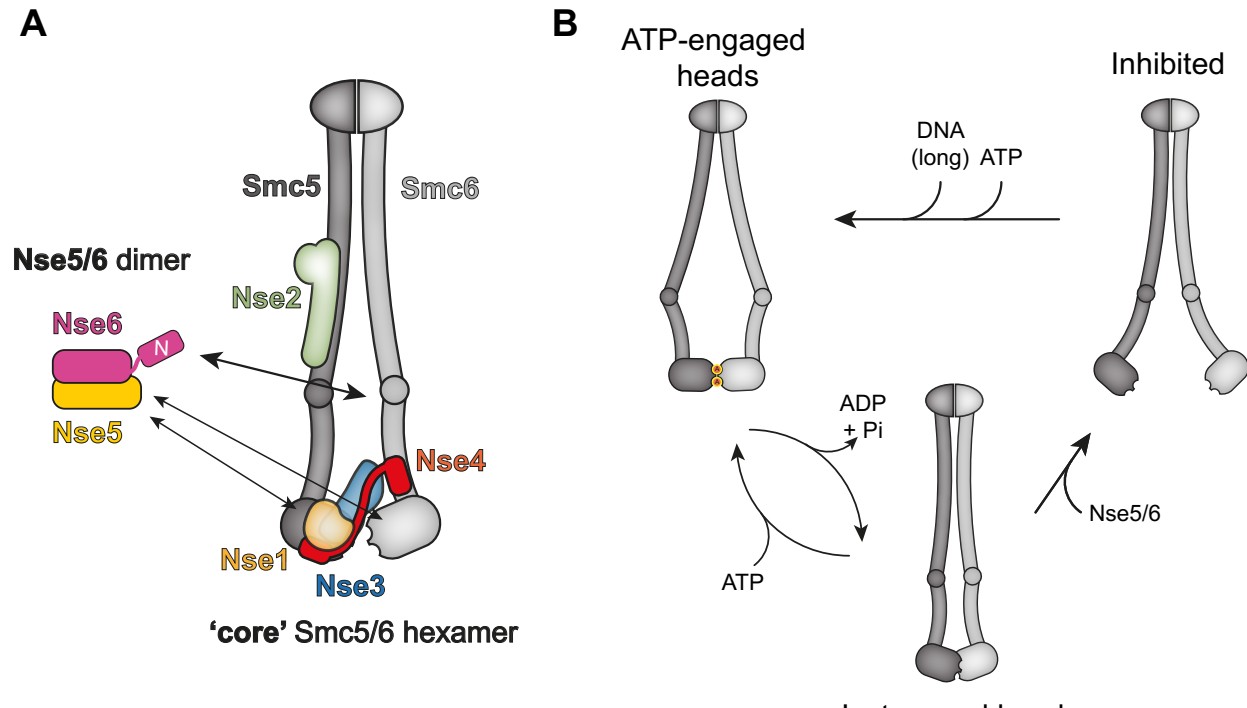

**Extended Data Fig. 1 | Schematics of Smc5/6 complex architecture. (a)** Overall architecture of the hexameric Smc5/6 core complex and the Nse5/6 dimer (adapted from[20]). For simplicity the Nse5/6 complex is shown separately with multiple arrows denoting various contact points with the core complex.

**(b)** Simplified schematics focusing only on changes in Smc5/6 dimer architecture during the ATPase cycle and upon binding to the Nse5/6 loader and the DNA substrate.

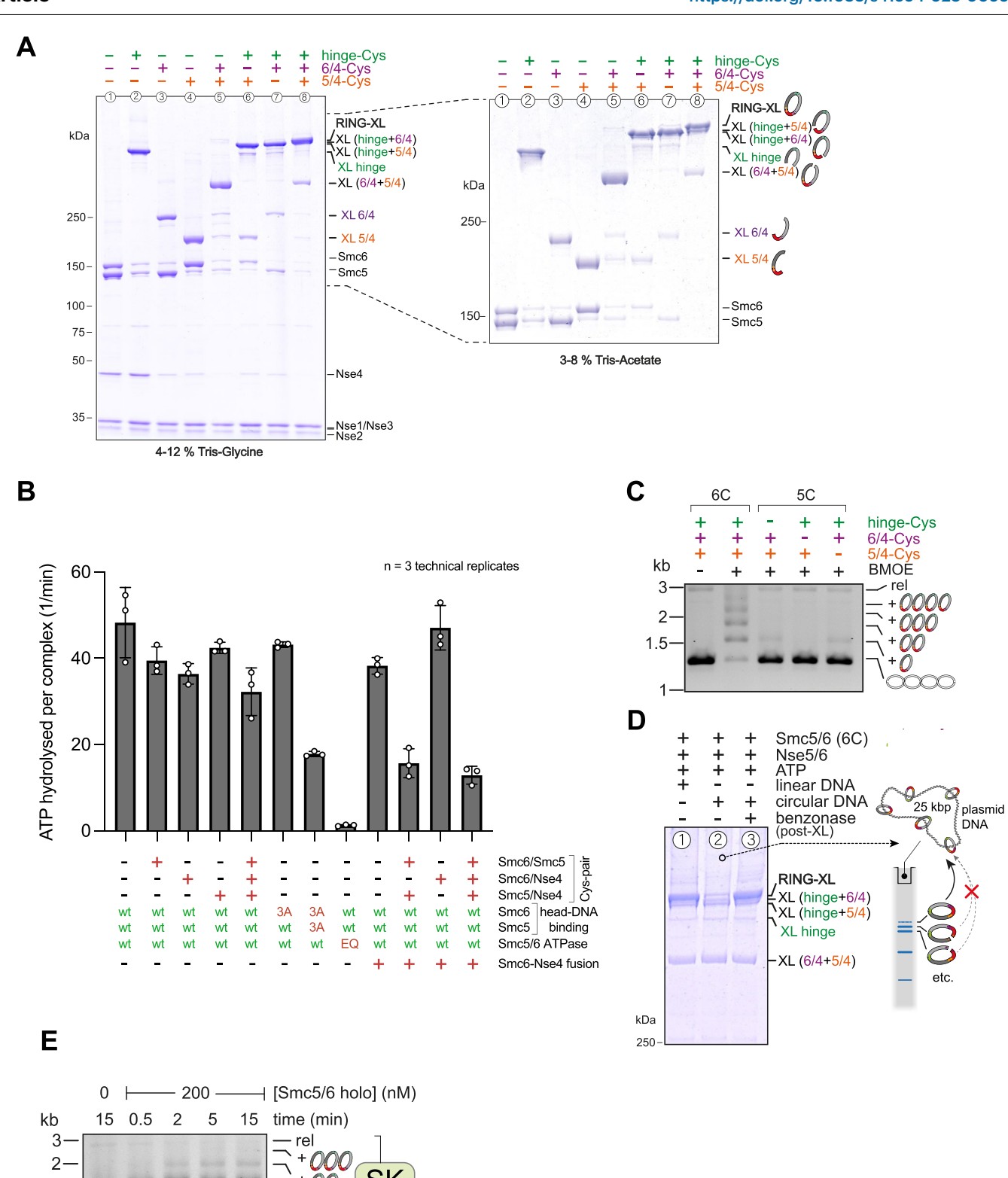

**Extended Data Fig. 2 | See next page for caption.**

**Extended Data Fig. 2 | Analysis of Smc5/6 complexes harboring multiple engineered cysteines. (a)** Crosslinking of Smc5/6 hexamers harboring multiple engineered cysteines as indicated. Identification of covalently closed ring species and intermediary crosslinking products by protein gel analysis on 2 types of gels (left: 4–12 % (w/v) gradient gel, right: 3–8 % (w/v) gradient gel) for proper separation of small and large proteins and detection by Coomassie staining. Note the presence of small amounts of ring species in two of the three control samples lacking one of the six cysteines (lanes 5 and 7), indicating minor off-target crosslinking. **(b)** ATPase activity assays with selected hexameric Smc5/6 complexes relevant for this study. Combinations of certain modifications lead to a reduction of ATPase activity. Error bars show standard deviations from the mean for three technical replicates. **(c)** Control experiments for entrapment experiments shown in Fig. 1 with preparations lacking a selected cysteine (5C). Note that low levels of co-entrapment detected with two of the three 5C samples are likely explained by weak off-target crosslinking. **(d)** Under conditions promoting DNA entrapment the ring species is visible in a protein gel in the presence of a linear (lane 1) but not a circular (lane 2) DNA substrate, presumably due to co-retention of the ring species with circular DNA in the loading well (see scheme on the right). The ring species re-appears after digestion of the circular substrate (lane 3). **(e)** Time-dependence of DNA entrapment in the SK ring (top panel) and the K compartment (bottom panel). Crosslinker was added to a sample aliquot at the indicated time points.

**A**

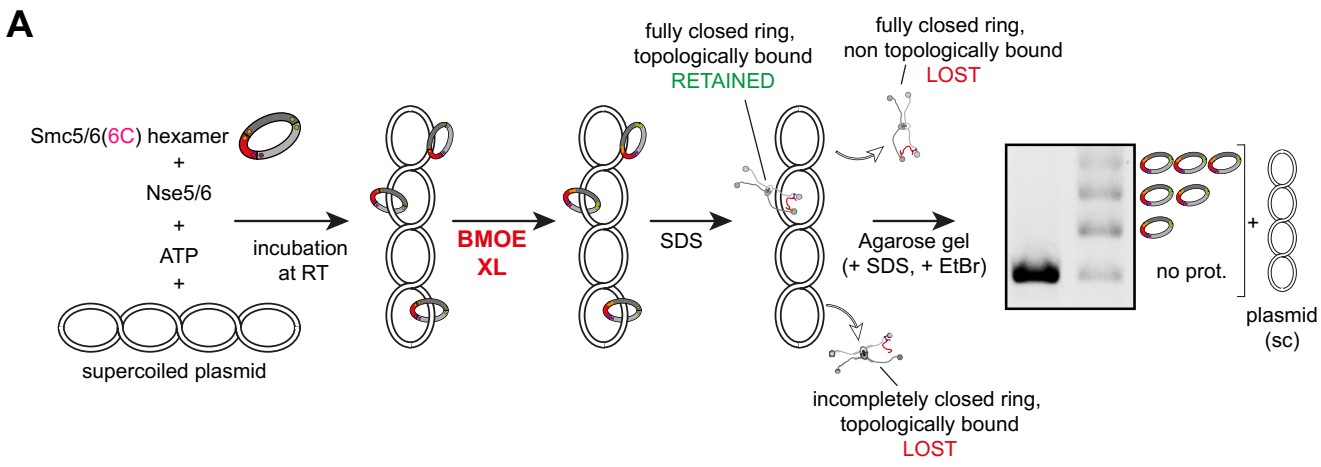

**B**

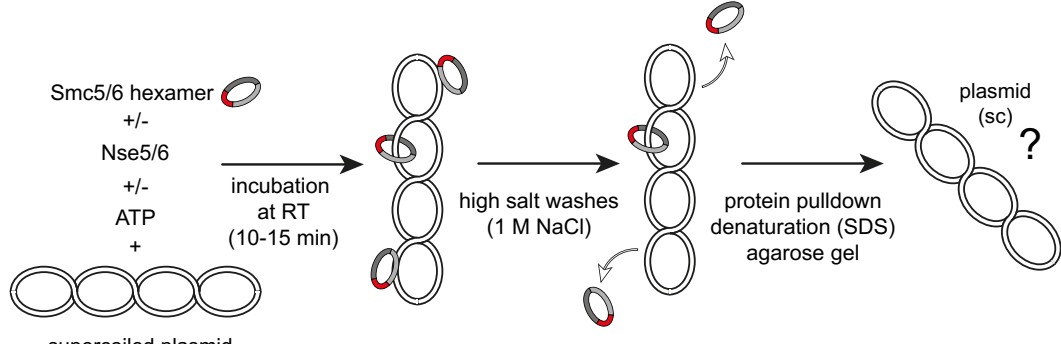

**Extended Data Fig. 3 | Schematic overview of assays used to examine the nature of Smc5/6 association with its DNA substrate. (a)** In the topological loading assay, a crosslinkable version of the Smc5/6 hexamer ('6C') is incubated with a small supercoiled ('sc') plasmid substrate in the absence or presence of ATP and Nse5/6. Upon crosslinking with BMOE and protein denaturation ('SDS') only fully crosslinked rings that were topologically associated with the substrate are retained, leading to a characteristic laddering pattern in agarose gels. **(b)** The salt-stable binding assay does not involve crosslinking. Complexes are first incubated with the substrate under low-salt conditions before the salt concentration is increased to 1 M NaCl by buffer changes. Smc5/6 complexes are immobilized on beads using a TwinStrep tag on Smc6, and co-purified plasmid substrate is examined on an agarose gel.

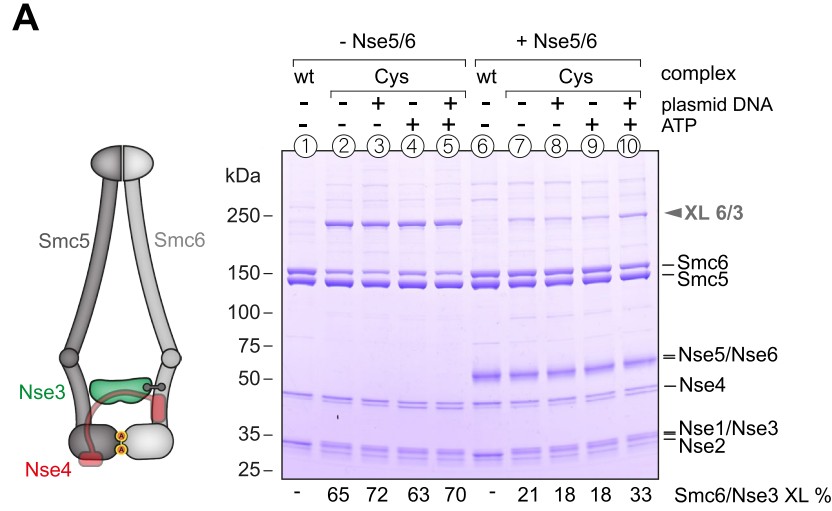

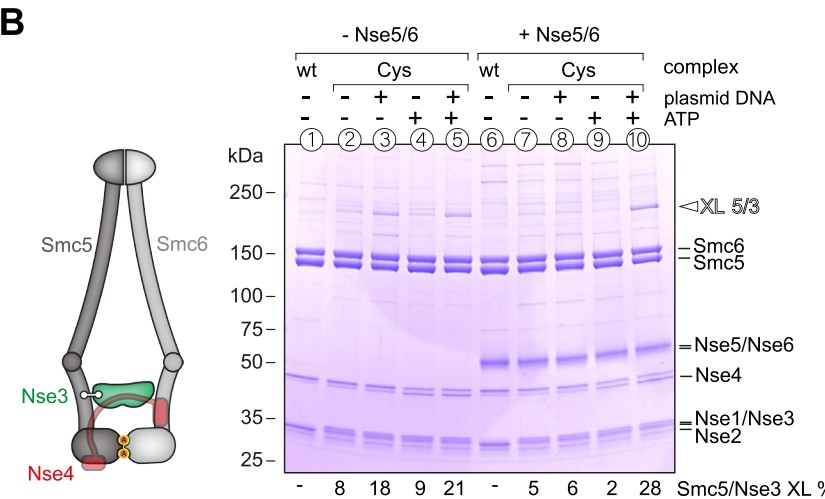

**Extended Data Fig. 4 | Analysis of Nse3 crosslinking to SMC arms.** Crosslinking of purified Smc5/6 hexamers with cysteines at the Smc6/Nse3 interface **(a)** or at the Smc5/Nse3 interface **(b)** in the presence and absence of ligands. Detection of crosslinked species by SDS-Page and Coomassie staining. Numbers below the gel indicate the efficiency of the respective crosslink.

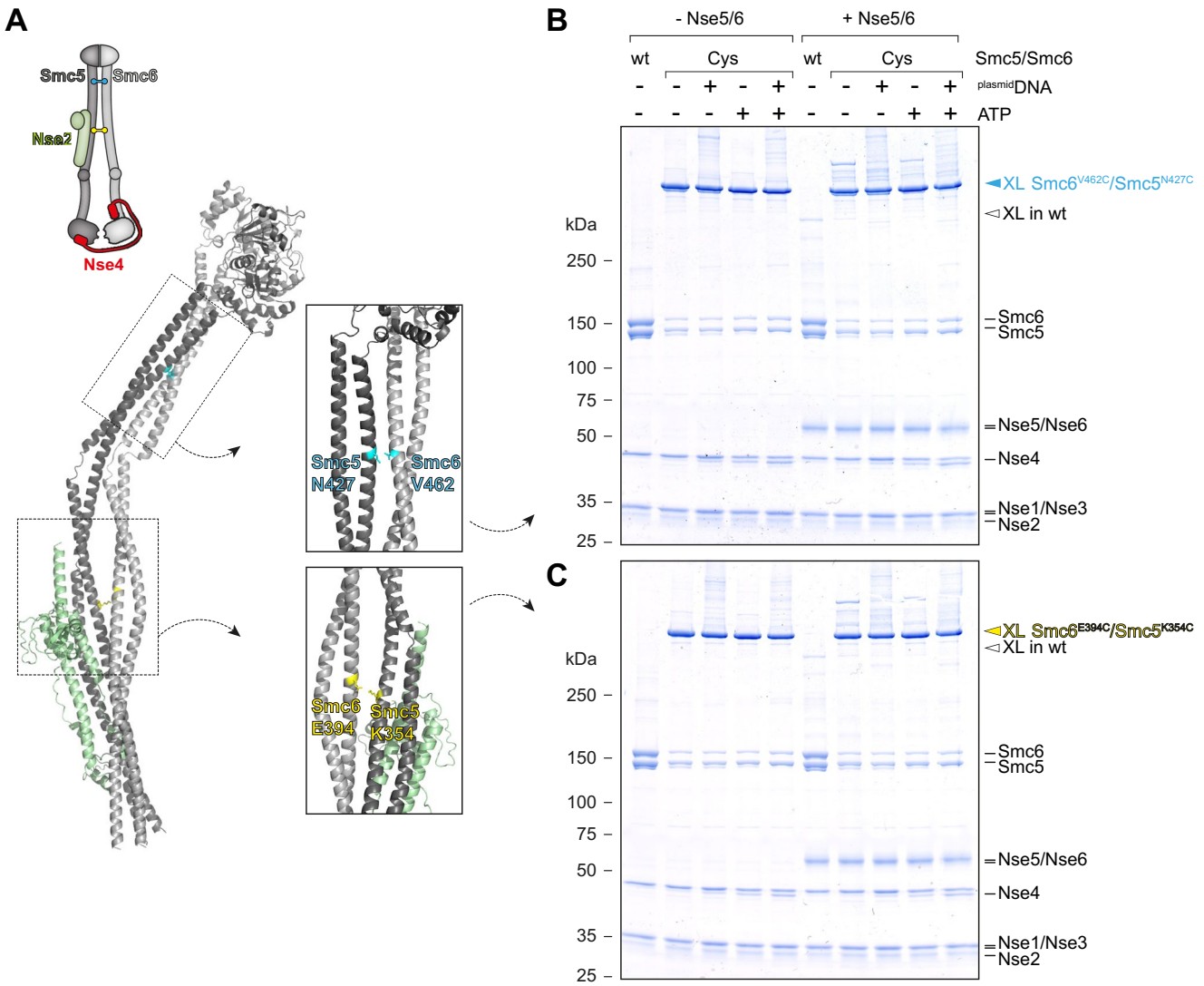

**Extended Data Fig. 5 | Head-distal Smc5/6 coiled-coil arms do not detectably open upon addition of Nse5/6, ATP, and/or DNA. a)** Model of an Smc5/Smc6/Nse2 complex obtained with AlphaFold-Multimer. The positions of two engineered cysteine pairs for inter-arm crosslinking are denoted. **(b, c)** Results from crosslinking experiments showing efficient arm alignment in all tested conditions. As in Fig. 3 and Supplementary Fig. 6.

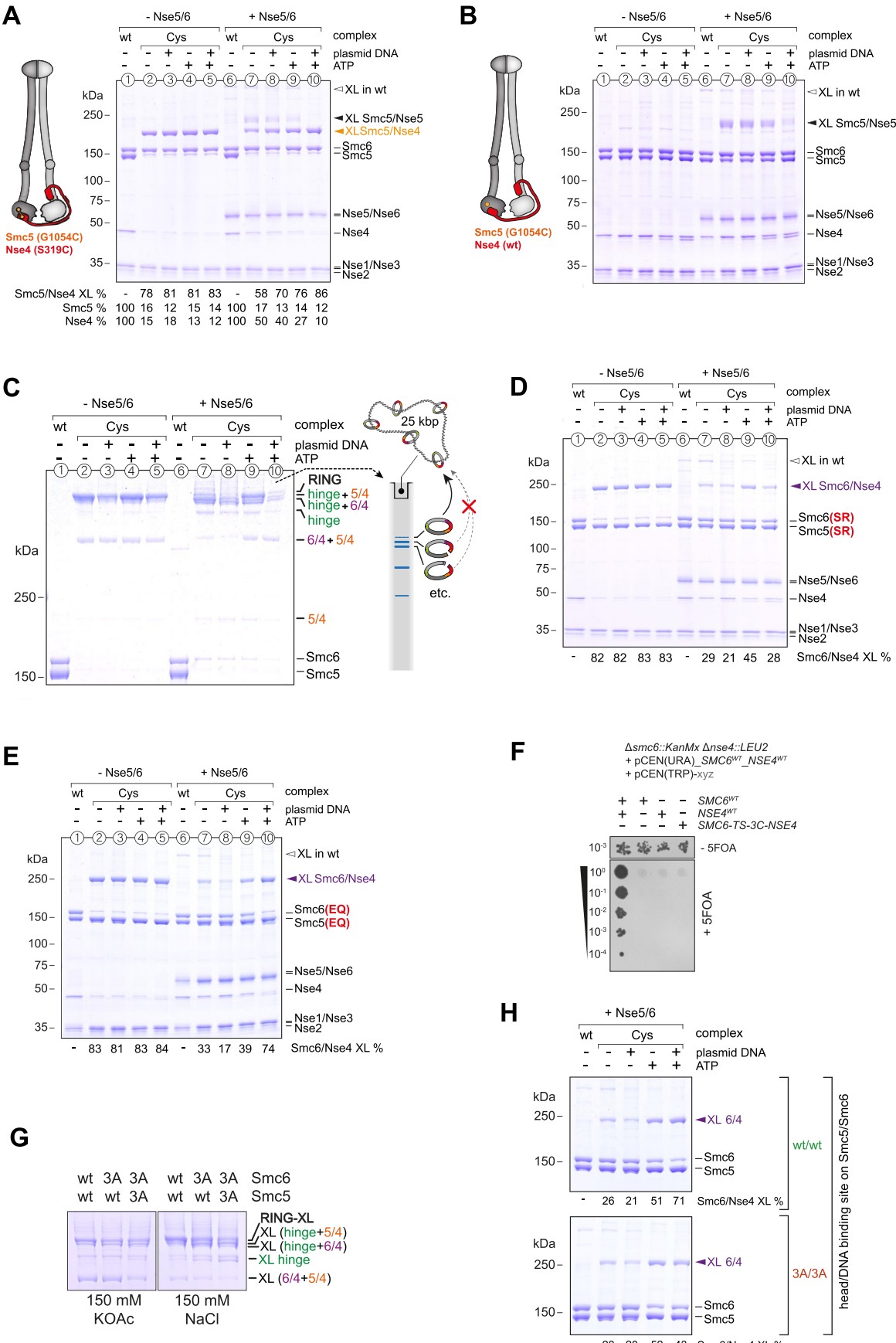

**Extended Data Fig. 6 | See next page for caption.**

**Extended Data Fig. 6 | Crosslinking of selected SMC/kleisin interfaces.**
Similar to Fig. 3 **(a)** The Smc5/Nse4 interface is efficiently crosslinked and does not detectably respond to the presence of ligands apart from weak off-target crosslinking between Smc5 and the loader subunit Nse5 [see (B)]. **(b)** Control reaction for (A) with protein samples lacking the engineered cysteine in Nse4 confirming off-target crosslinking of Smc5(G1054C) to Nse5. **(c)** Addition of the loader reduces abundance of the ring species due to gate opening. The pattern in the presence of ligands (ATP and plasmid DNA) mirrors the one obtained with the Smc6/Nse4 interface (Fig. 3), except when loader, ATP and plasmid are added, presumably due to co-retention of ring species with circular DNA in the loading well (see scheme on the right). **(d)** As in Fig. 3 but with Smc5 and Smc6 subunits

carrying the signature motif head-engagement mutation ('SR'). **(e)** As in Fig. 3 but with Smc5 and Smc6 subunits carrying the Walker B ATP hydrolysis mutation ('EQ'). **(f)** Fusion of Smc6 to Nse4 with the linker used in our in vitro assays is lethal in yeast. Plasmid shuffling assay as in Supplementary Fig. 7f, but with a deletion mutant for both *smc6* and *nse4*. Adding back both wildtype genes separately, but not either of them alone or a fused version, restores viability. **(g)** Effect of mutating the head DNA binding interfaces on Smc5 and/or Smc6 on formation of a closed SMC/Kleisin (SK) ring. **(h)** Effect of mutating the head DNA binding interfaces on Smc5 and Smc6 on Smc6/Nse4 gate closure in the presence of the loader and various ligands.

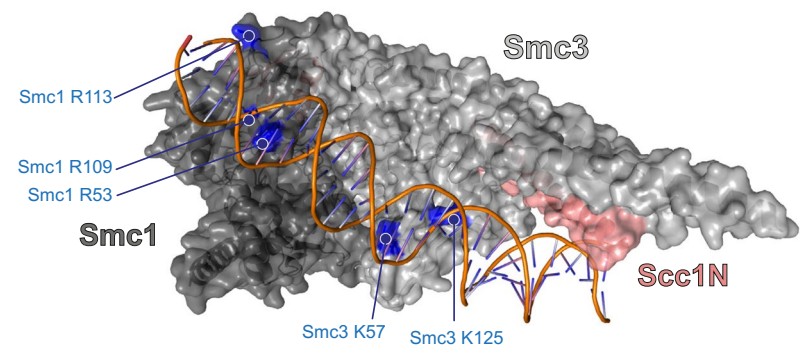

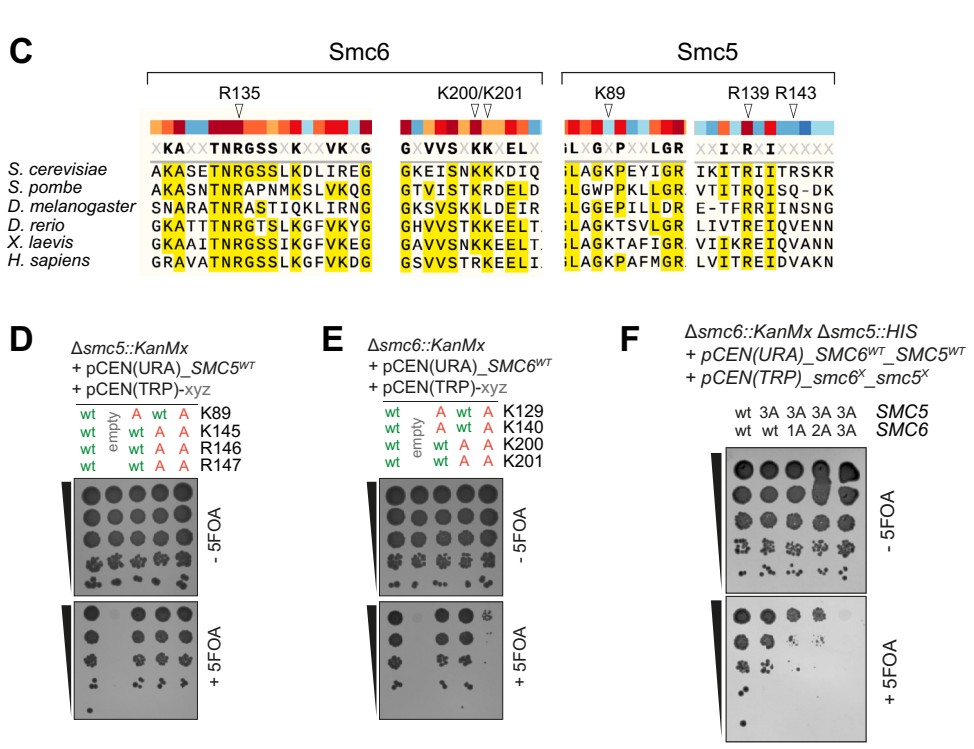

**Extended Data Fig. 7 | See next page for caption.**

**Extended Data Fig. 7 | Identification of putative DNA binding residues on Smc6 and Smc5. (a)** Model of ATP-engaged Smc6/Smc5 heads with the Nse4 N-terminus bound to the Smc6 neck. AlphaFold-Multimer models of Smc6/Nse4 and Smc5 were superimposed on their counterparts of the cohesin complex (PDB: 6ZZ6, see panel B). The DNA molecule from the cohesin cryo-EM structure contacts several putative DNA binding residues on Smc6 and Smc5. **(b)** Similar view on top of engaged heads in the cohesin cryo-EM structure (PDB: 6ZZ6) with DNA-interacting residues on Smc3 and Smc1 indicated. Note that the Scc2 molecule is not shown for simplicity. **(c)** Sequence alignment showing strong evolutionary conservation of examined residues in Smc6. Smc5 residues show weaker conservation consistent with results of functional assays shown in Fig. 4. **(d)** Positively charged residues on the Smc5 head were mutated to alanine ('A') and the mutant alleles were tested for function by plasmid shuffling. As in Fig. 4b but with residues selected based on a recent cryo-EM structure of Smc5/6 (Yu *et al.*[43]). **(e)** As in (D) but for residues on the Smc6 head. **(f)** Positively charged residues on Smc5 and Smc6 heads were mutated to alanines in combinations. As in Fig. 4b and c.

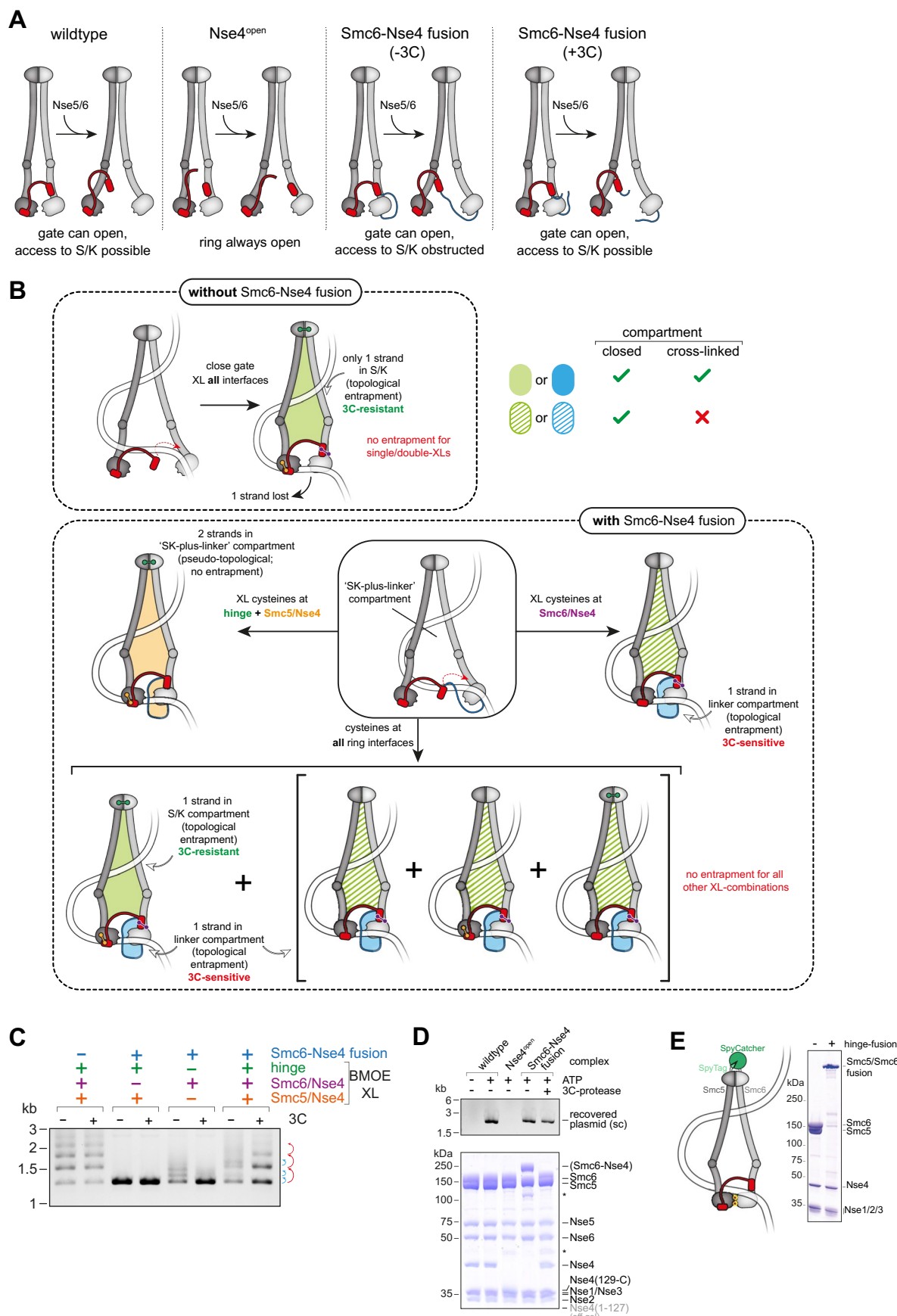

**Extended Data Fig. 8 | See next page for caption.**

**Extended Data Fig. 8 | Entrapment assays involving Smc6-Nse4 fusion and BMOE crosslinking of interfaces. (a)** Schematic representation of complexes used for experiments in Fig. 5 and Supplementary Fig. 8d. The 'Nse4$^{open}$' complex has a split kleisin and thus has its SK ring permanently opened. The 'Smc6-Nse4 fusion' complex contains a peptide linker fusing the C-terminus of Smc6 with the N-terminus of Nse4. The linker contains a recognition site for the HRV 3C protease and can thus be opened by cleavage. **(b)** Schematic overview of entrapment in the absence and presence of the Smc6-Nse4 fusion. Without the fusion (top) during initial DNA segment capture the Smc6/Nse4 gate is opened by Nse5/6 (Smc5 and Smc6 arm distance is exaggerated for clarity). Upon gate closure and ATP hydrolysis, one of the loop strands is lost after escape between the disengaged heads, and the other strand becomes topologically entrapped in the SK ring. The green-filled area indicates the lumen of the ring compartment (SK) which is maintained even after protein denaturation due to crosslinking. (bottom) Scenarios for DNA entrapment by complexes with Smc6-Nse4 fusion (top row, middle panel) in combination with different cysteine pairs for crosslinking (other panels). Top row, left panel: Upon crosslinking of Smc5/Smc6 and Smc5/Nse4 interfaces an SK-plus-linker joint compartment becomes denaturation-resistant (orange-filled area) leading to DNA loop entrapment

(not detected in this assay) rather than DNA entrapment. Top row, right panel: Crosslinking of only the Smc6/Nse4 interface leads to a denaturation-resistant linker compartment (blue-filled area) that topologically entraps a DNA strand. The SK compartment is also formed and entraps the other DNA strand, but it is sensitive to denaturation (green-dashed area). Bottom row: Crosslinking of all three interfaces entraps one DNA strand in a denaturation-resistant SK compartment (green-filled area) and another in the linker compartment (blue-filled area), only the latter of which can be released by incubation with 3C protease. Incompletely crosslinked rings lacking the Smc5/Smc6 or Smc5/Nse4 crosslinks (or both) entrap only the DNA strand in the linker compartment (in square bracket). **(c)** As in Fig. 5c but with cleavage of the linker prior to mixing of samples for DNA entrapment. **(d)** Salt-stable DNA binding of Smc5/6 complexes with a split Nse4 protein ('Nse4 open') or linked Smc6 and Nse4 proteins ('Smc6-Nse4' fusion protein). As in Fig. 3d. Pre-treatment with 3C protease cleaves the fusion linker peptide but does not alter DNA binding. Asterisks denote unspecific degradation products. **(e)** Insertion of a Spy-tag and Spy-Catcher into the Smc5 and Smc6 hinge-domains (see scheme on the left), respectively, leads to permanent, covalent fusion of the hinge domains as shown in the protein gel on the right.

# Reporting Summary

## Statistics

For all statistical analyses, confirm that the following items are present in the figure legend, table legend, main text, or Methods section.

| n/a | Confirmed | |
|---|---|---|
| ☐ | ☒ | The exact sample size ($n$) for each experimental group/condition, given as a discrete number and unit of measurement |
| ☐ | ☒ | A statement on whether measurements were taken from distinct samples or whether the same sample was measured repeatedly |
| ☒ | ☐ | The statistical test(s) used AND whether they are one- or two-sided<br>*Only common tests should be described solely by name; describe more complex techniques in the Methods section.* |
| ☒ | ☐ | A description of all covariates tested |
| ☒ | ☐ | A description of any assumptions or corrections, such as tests of normality and adjustment for multiple comparisons |
| ☐ | ☒ | A full description of the statistical parameters including central tendency (e.g. means) or other basic estimates (e.g. regression coefficient) AND variation (e.g. standard deviation) or associated estimates of uncertainty (e.g. confidence intervals) |
| ☒ | ☐ | For null hypothesis testing, the test statistic (e.g. $F$, $t$, $r$) with confidence intervals, effect sizes, degrees of freedom and $P$ value noted<br>*Give P values as exact values whenever suitable.* |
| ☒ | ☐ | For Bayesian analysis, information on the choice of priors and Markov chain Monte Carlo settings |
| ☒ | ☐ | For hierarchical and complex designs, identification of the appropriate level for tests and full reporting of outcomes |
| ☒ | ☐ | Estimates of effect sizes (e.g. Cohen's $d$, Pearson's $r$), indicating how they were calculated |

*Our web collection on statistics for biologists contains articles on many of the points above.*

## Software and code

Policy information about availability of computer code

| Data collection | VisionWorks 11.1 (Agarose gel images); Amersham Typhoon 3.0.0.2 (Southern blots); |
|---|---|
| Data analysis | Fiji ImageJ 2.1.0/1.53c; Prism for GraphPad 9.0; |

For manuscripts utilizing custom algorithms or software that are central to the research but not yet described in published literature, software must be made available to editors and reviewers. We strongly encourage code deposition in a community repository (e.g. GitHub). See the Nature Portfolio guidelines for submitting code & software for further information.

## Data

Policy information about availability of data

All manuscripts must include a data availability statement. This statement should provide the following information, where applicable:
- Accession codes, unique identifiers, or web links for publicly available datasets
- A description of any restrictions on data availability
- For clinical datasets or third party data, please ensure that the statement adheres to our policy

All raw data are included as Source Data. The following publicly available datasets were used: PDB: 7TVE (yeast Smc5/6 ATP DNA); PDB: 6ZZ6 (yeast cohesin-Scc2-DNA).

# Human research participants

Policy information about studies involving human research participants and Sex and Gender in Research.

| | |
|---|---|
| Reporting on sex and gender | n.a. |
| Population characteristics | n.a. |
| Recruitment | n.a. |
| Ethics oversight | n.a. |

Note that full information on the approval of the study protocol must also be provided in the manuscript.

# Field-specific reporting

Please select the one below that is the best fit for your research. If you are not sure, read the appropriate sections before making your selection.

☒ Life sciences  ☐ Behavioural & social sciences  ☐ Ecological, evolutionary & environmental sciences

For a reference copy of the document with all sections, see nature.com/documents/nr-reporting-summary-flat.pdf

# Life sciences study design

All studies must disclose on these points even when the disclosure is negative.

| | |
|---|---|
| Sample size | Sample size was chosen to permit robust detection by gel analysis. |
| Data exclusions | No data was excluded from the analyses. |
| Replication | Quantitative measurements (ATP hydrolysis rates) were performed in three technical replicates. Means and standard deviation are reported. All other experiments were repeated at least three times from independently grown cultures and purified proteins with comparable outcomes. |
| Randomization | Experiments did not involve the allocation of groups and randomization was not relevant. |
| Blinding | Experiments did not involve the allocation of groups and blinding was not relevant. |

# Reporting for specific materials, systems and methods

We require information from authors about some types of materials, experimental systems and methods used in many studies. Here, indicate whether each material, system or method listed is relevant to your study. If you are not sure if a list item applies to your research, read the appropriate section before selecting a response.

## Materials & experimental systems

| n/a | Involved in the study |
|---|---|
| ☐ | ☒ Antibodies |
| ☒ | ☐ Eukaryotic cell lines |
| ☒ | ☐ Palaeontology and archaeology |
| ☒ | ☐ Animals and other organisms |
| ☒ | ☐ Clinical data |
| ☒ | ☐ Dual use research of concern |

## Methods

| n/a | Involved in the study |
|---|---|
| ☒ | ☐ ChIP-seq |
| ☒ | ☐ Flow cytometry |
| ☒ | ☐ MRI-based neuroimaging |

## Antibodies

| | |
|---|---|
| Antibodies used | The mouse monoclonal anti-V5 antibody (BioRad) was used (10 µg per sample) for immuno-precipitation of Pk6 epitope tagged Smc5 proteins. |
| Validation | Specificity of immuno-precipitation was validated using control sample lacking cysteine for cross-linking. |

