## [Peer Review File · Nature Structural & Molecular Biology]

Peer Review Information

Journal: Structural & Molecular Biology

Manuscript Title: DNA segment capture by Smc5/6 holo-complexes

Corresponding author name(s): Stephen Gruber

Editorial Notes:

Transferred manuscripts This manuscript has been previously reviewed at another journal that is not operating a transparent peer review scheme. This document only contains reviewer comments, rebuttal and decision letters for versions considered at Nature Structural & Molecular Biology.

Reviewer Comments & Decisions:

Decision Letter, initial version:
--

Message: Our ref: NSMB-A46927-T

23rd Jan 2023

Dear Dr. Gruber,

Thank you for submitting your revised manuscript "DNA segment capture by Smc5/6 holo-complexes." (NSMB-A46927-T). We sincerely apologize for the delay in responding, which was due to our editorial team having been short-staffed in the last months together with the longer waiting periods over the holidays. Your manuscript has now been seen by the original referees and their comments are below. The reviewers find that the paper has improved in revision, and therefore we'll be happy in principle to publish it in Nature Structural & Molecular Biology, pending minor revisions to satisfy the referees' final requests and to comply with our editorial and formatting guidelines.

We are now performing detailed checks on your paper and will send you a checklist detailing our editorial and formatting requirements in the next couple of weeks. Please do not upload the final materials and make any revisions until you receive this additional information from us.

To facilitate our work at this stage, we would appreciate if you could send us the main text as a word file. Please make sure to copy the NSMB account (cc'ed above).

Sincerely,
Sara

Sara Osman, Ph.D.
Associate Editor
Nature Structural & Molecular Biology

Reviewer #1 (Remarks to the Author):

I have re-read the revised manuscript by Taschner and Gruber and the authors' response. The authors now answered all my questions and improved the manuscript so that now it's more clearly described which compartment entraps DNAs and easier to understand the results. They added new results showing that DNA laddering was disappeared with linearized DNA (Figure 1C) and cross linking efficiency (Figure S6G and H). Although my original concern was biological relevance of the segment capture model, I understand that the focus of this manuscript is topological loading mechanism and additional functions including translocation or loop extrusion will be addressed in the future, that I'm very much anticipating. Now I would recommend this study for the publication in Nature Structural & Molecular Biology.

Reviewer #2 (Remarks to the Author):

The revised manuscript by Taschner and Gruber has improved greatly. The authors have addressed all of my points. The paper has become easier to read, and the exciting message of the paper comes across well. I congratulate the authors with this beautiful work, and I can highly recommend publication.

I notice only one minor 'typo', which is the mis-formatting of the Serrano reference on page 8.

Author Rebuttal to Initial comments

Reviewer #1 (Remarks to the Author):

I have re-read the revised manuscript by Taschner and Gruber and the authors' response. The authors now answered all my questions and improved the manuscript so that now it's more clearly described which compartment entraps DNAs and easier to understand the results. They added new results showing that DNA laddering was disappeared with linearized DNA (Figure 1C) and cross linking efficiency (Figure S6G and H). Although my original concern was biological relevance of the segment capture model, I understand that the focus of this manuscript is topological loading mechanism and additional

functions including translocation or loop extrusion will be addressed in the future, that I'm very much anticipating. Now I would recommend this study for the publication in Nature Structural & Molecular Biology.

We thank the reviewer for the supportive comments.

Reviewer #2 (Remarks to the Author):

The revised manuscript by Taschner and Gruber has improved greatly. The authors have addressed all of my points. The paper has become easier to read, and the exciting message of the paper comes across well. I congratulate the authors with this beautiful work, and I can highly recommend publication.

Many thanks for the open appreciation of the work.

I notice only one minor 'typo', which is the mis-formatting of the Serrano reference on page 8.

The mis-formatted reference has been corrected.

Final Decision Letter:

Message 1st Mar 2023

:

Dear Dr. Gruber,

We are now happy to accept your revised paper "DNA segment capture by Smc5/6 holo-complexes" for publication as a Article in Nature Structural & Molecular Biology.

After the grant of rights is completed, you will receive a link to your electronic proof via email with a request to make any corrections within 48 hours. If, when you receive your proof, you cannot meet this deadline, please inform us at

rjsproduction@springernature.com immediately.

As soon as your article is published, you can generate your shareable link by entering the DOI of your article here: http://authors.springernature.com/share. Corresponding authors will also receive an automated email with the shareable link

Your paper will be published online soon after we receive proof corrections and will appear in print in the next available issue. You can find out your date of online publication by contacting the production team shortly after sending your proof corrections. Content is published online weekly on Mondays and Thursdays, and the embargo is set at 16:00 London time (GMT)/11:00 am US Eastern time (EST) on the day of publication. Now is the time to inform your Public Relations or Press Office about your paper, as they might be interested in promoting its publication. This will allow them time to prepare an accurate and satisfactory press release. Include your manuscript tracking number (NSMB-A46927A) and our journal name, which they will need when they contact our press office.

About one week before your paper is published online, we shall be distributing a press release to news organizations worldwide, which may very well include details of your work. We are happy for your institution or funding agency to prepare its own press release, but it must mention the embargo date and Nature Structural & Molecular Biology. If you or your Press Office have any enquiries in the meantime, please contact press@nature.com.

If you have not already done so, we strongly recommend that you upload the step-by-step protocols used in this manuscript to the Protocol Exchange. Protocol Exchange is an open online resource that allows researchers to share their detailed experimental know-how. All uploaded protocols are made freely available, assigned DOIs for ease of citation and fully searchable through nature.com. Protocols can be linked to any publications in which they are used and will be linked to from your article. You can also establish a dedicated page to collect all your lab Protocols. By uploading your Protocols to Protocol Exchange, you are

enabling researchers to more readily reproduce or adapt the methodology you use, as well as increasing the visibility of your protocols and papers. Upload your Protocols at www.nature.com/protocolexchange/. Further information can be found at www.nature.com/protocolexchange/about.

Please note that *Nature Structural & Molecular Biology* is a Transformative Journal (TJ). Authors may publish their research with us through the traditional subscription access route or make their paper immediately open access through payment of an article-processing charge (APC). Authors will not be required to make a final decision about access to their article until it has been accepted. [Find out more about Transformative Journals](https://www.springernature.com/gp/open-research/transformative-journals)

Authors may need to take specific actions to achieve [compliance with funder and institutional open access mandates](https://www.springernature.com/gp/open-research/funding/policy-compliance-faqs). If your research is supported by a funder that requires immediate open access (e.g. according to [Plan S principles](https://www.springernature.com/gp/open-research/plan-s-compliance)) then you should select the gold OA route, and we will direct you to the compliant route where possible. For authors selecting the subscription publication route, the journal's standard licensing terms will need to be accepted, including [self-archiving policies](https://www.springernature.com/gp/open-research/policies/journal-policies). Those licensing terms will supersede any other terms that the author or any third party may assert apply to any version of the manuscript.

Sincerely,

Dimitris Typas
Associate Editor
Nature Structural & Molecular Biology
ORCID: 0000-0002-8737-1319
